# Integrative reconstruction of cancer genome karyotypes using InfoGenomeR

Yeonghun Lee [1] & Hyunju Lee [1✉]

Annotation of structural variations (SVs) and base-level karyotyping in cancer cells remains challenging. Here, we present Integrative Framework for Genome Reconstruction (InfoGenomeR)-a graph-based framework that can reconstruct individual SVs into karyotypes based on whole-genome sequencing data, by integrating SVs, total copy number alterations, allele-specific copy numbers, and haplotype information. Using whole-genome sequencing data sets of patients with breast cancer, glioblastoma multiforme, and ovarian cancer, we demonstrate the analytical potential of InfoGenomeR. We identify recurrent derivative chromosomes derived from chromosomes 11 and 17 in breast cancer samples, with homogeneously staining regions for *CCND1* and *ERBB2*, and double minutes and breakage-fusion-bridge cycles in glioblastoma multiforme and ovarian cancer samples, respectively. Moreover, we show that InfoGenomeR can discriminate private and shared SVs between primary and metastatic cancer sites that could contribute to tumour evolution. These findings indicate that InfoGenomeR can guide targeted therapies by unravelling cancer-specific SVs on a genome-wide scale.

[1] School of Electrical Engineering and Computer Science, Gwangju Institute of Science and Technology, Gwangju, South Korea. ✉email: hyunjulee@gist.ac.kr

Cancer cells acquire numerous changes in their DNA, ranging from point mutations to DNA rearrangements, that ultimately result in a complex cancer-associated genome. Recurrent chromosomal structural variations (SVs) have been linked to tumorigenesis, including simple SVs such as tandem duplications, deletions, inversions and insertions, which have been extensively studied[1,2], as well as more complex SVs such as translocations, fold-back inversions, chromothripsis, homogeneously staining regions (HSRs, representing repetitive gene amplification) and double minutes (DMs, extra-chromosomal DNA)[3,4]. Traditional karyotyping techniques, such as G-banding and fluorescent in situ hybridisation (FISH) can reveal the presence of complex SVs in derivative chromosomes (by-product of the recombination of multiple chromosomes with intact centromeres) or marker chromosomes (abnormal chromosomes with unidentified genomic segments)[5]. However, owing to their limited resolution (~5 Mb), standard karyotyping techniques cannot be used to accurately identify complex SVs in derivative or marker chromosomes.

High-throughput sequencing has advanced our understanding of SVs by resolving the genomic changes at the single-base level. Early-stage methods have been developed to detect SVs using discordant and split reads from sequencing data[6–9]; however, these methods have limited detection ability for SV breakpoints in local genomic windows. Recently, several methods[10–19] that integrate genomic information, such as cancer purity and ploidy, total copy number alterations (CNAs), allele-specific CNAs and haplotype information, have been developed to identify SVs. They use a graph-based representation for rearranged cancer genomes but do not analyse the actual karyotypes of linear and/or circular chromosomes, thus, not producing karyotypic topologies such as HSRs, DMs, or chromothripsis. Global reconstruction of genome karyotypes in cancers may allow uncovering of the mechanism underlying cancer development and evolution.

In this article, we present a method to reconstruct cancer genome karyotypes based on complex topology analysis, providing a haplotype graph-based representation. Our graph-based framework, named Integrative Framework for Genome Reconstruction (InfoGenomeR), uses a breakpoint graph to model the connectivity among genomic segments on a genome-wide scale using as input SV calls, unmapped reads, read-depth information and single nucleotide polymorphisms (SNPs). Furthermore, the InfoGenomeR tool classifies the rearrangement topologies and derives the cancer genome karyotypes from the haplotype graphical output (Supplementary Fig. 1). We show the analytical potential of our method by comparing it with existing tools using simulation data and cancer cell line data. Moreover, using WGS data from The Cancer Genome Atlas (TCGA)[20–22] and European Genome–phenome Archive (EGA)[23], we show that InfoGenomeR can reconstruct the karyotypes of cancer cells and distinguish between private and shared SVs in primary and metastatic cancer cells, and reveal tumour evolution.

## Results

**InfoGenomeR reconstructs candidate genome karyotypes.** First, InfoGenomeR evaluates all reads in WGS data sets, generates initial SV calls using the tools DELLY2[6], Manta[7] and novoBreak[8] (Fig. 1a), and performs initial CN segmentation using BIC-seq2[24]. Then, it constructs an initial breakpoint graph of local genomic segments using the initial SV and CN breakpoints. The breakpoint graph is composed of nodes and segment edges, reference edges, and SV edges. The following three-step iterations update the initial breakpoint graph. In each iteration, (i) local genomic segments are refined, (ii) integer CNs of genomic segments are

estimated using purity and ploidy (ABSOLUTE[25]) and (iii) the integer programming of the CN balance condition[26] determines the edge multiplicities of the breakpoint graph and removes zero-multiplicity SVs. Each iteration restarts with the SV set without zero-multiplicity SVs, CN segmentation is performed without the previous false-positive SV breakpoints, and integer CNs of segments are recalculated. Iterations are performed until the graph converges (no zero-multiplicity SV is observed). The iterations are composed of first and second rounds of iterations depending on the segmentation parameter, and the CN segments are merged with their neighbour CN segments more commonly in the second-round iterations than in the first-round iterations. At the intermediate step between the first and second rounds of iterations, the discordant or unmapped reads, which do not pair properly, are remapped to the sequences of candidate adjacencies from unbalanced nodes. (Fig. 1b). Then, candidate adjacencies supported by their reads are generated, and the second-round iterations finalise the breakpoint graph. Next, integer CNs are divided into ASCNs using negative binomial models for the different depths of heterozygous SNPs, and the expectation–maximisation (EM) algorithm is used for estimating parameters. Integer programming under the CN balance condition with the ASCNs constructs the allele-specific breakpoint graph and then the imbalanced heterozygous SNP sequences are phased (Fig. 1c). Genomic segments with balanced heterozygous SNPs are phased using a hidden Markov model (BEAGLE[27]), and the final haplotype breakpoint graph is constructed (Fig. 1d). Eulerian paths can be enumerated to obtain candidate genomes by pairing breakpoint graph edges using a multiway tree structure[28] with minimum-entropy search. In the end, InfoGenomeR generates candidate karyotypes of the cancer cells at the haplotype level (Fig. 1e).

**InfoGenomeR outperforms other variant-calling methods.** Based on the simulated data sets (Supplementary Note 1), we evaluated the performance of InfoGenomeR against eight other tools in six restricted variant-calling categories, SVs, SV copy numbers (SVCNs), CNA breakpoints, integer CNs, ASCNs and haplotype. These six categories were evaluated for the total and somatic mode. Different methods were compared to detect variants in each category. The performance of the individual methods was evaluated before integrating them into InfoGenomeR. We performed fourfold cross-validation for each haplotype coverage, where the selected parameters or thresholds were determined by enumerating a defined range of values (Supplementary Table 1). To compare InfoGenomeR with JaBbA, which is the recent graph SV caller, we ran JaBbA[16] using the same SV union set input (DELLY2, Manta and novoBreak) that we used for InfoGenomeR. Because JaBbA was sensitive to the input purity and ploidy hyperparameter, we used the purity and ploidy estimation of InfoGenomeR for the JaBbA input. We tested various hyperparameter settings for JaBbA along with the JaBbA recommendation and selected the best setting for SV detection (Supplementary Fig. 2). We judged the performance metrics on the precision and recall of each variant-calling category (Supplementary Table 2).

InfoGenomeR achieved the highest total (precision, 0.987; recall, 0.825) and somatic (precision, 0.981; recall, 0.919) SV calling performance, at a haplotype coverage of 15X, when compared with the three methods using discordant and split reads (DELLY2, Manta, and novoBreak), and the three methods using both discordant/split reads and read depths (CONSERTING[9,10], Weaver[11] and JaBbA) (Fig. 2). Furthermore, JaBbA produced second-best results for SV calling. Our results showed that the integrative strategy of InfoGenomeR imparted an enhanced performance over individual SV tools (DELLY2, Manta and

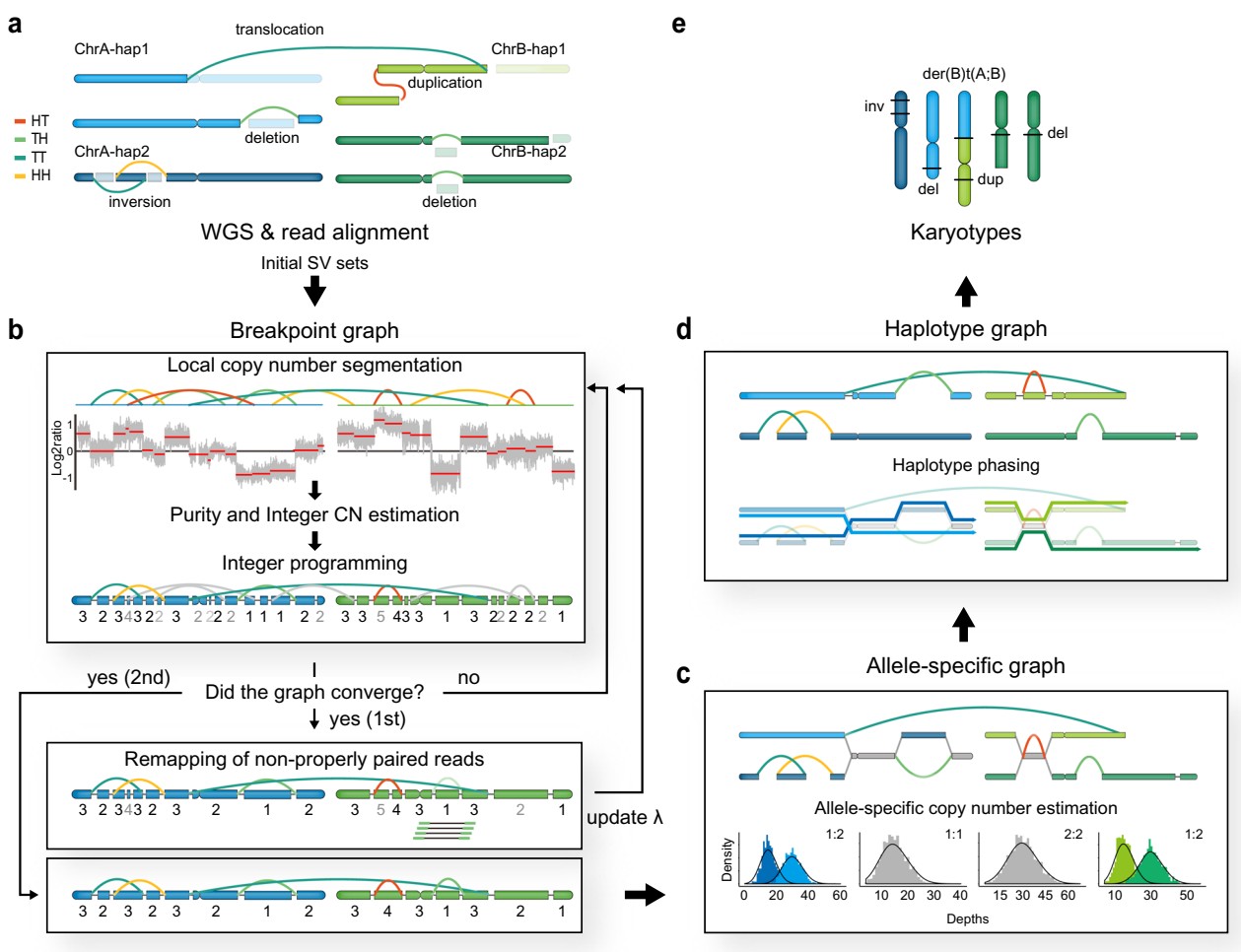

**Fig. 1 Schematic diagram of InfoGenomeR steps. a** Each of the two chromosomes (chrA and chrB) has two haplotypes (hap1 and hap2), with head-to-tail (HT), tail-to-head (TH), tail-to-tail (TT) and head-to-head (HH) SVs. **b** Breakpoint graph construction. After reads from whole-genome sequencing data (WGS) are aligned, a breakpoint graph that is composed of nodes and segment edges (blue and green boxes), reference edges (black lines) and SV edges (coloured lines) is constructed. An iteration consisting of three steps is shown in the first box. The $\log_2$ratio represents a normalised copy number (CN), and numbers under the nodes are integer CNs (low-confidence CNs are shown in grey). Note that the false-positive edges change to grey edges after integer programming. Once the graph converges (first iterations), reads supporting a deletion (green) are remapped, and the segmentation parameter $\lambda$ is updated before second iterations. **c** Allele-specific graph construction. ASCNs are measured for each segment using a negative binomial model, and imbalanced segments are divided into allelic segments (light blue and blue for chrA, light green and green for chrB). Balanced segments (grey) remain in the same states in the breakpoint graph. **d** Haplotype graph construction. Allelic segments are phased into haplotypes using the HMM (blue and green arrows) to construct the haplotype graph. **e** The Eulerian path-finding problem in the haplotype graph finally reconstructs the cancer genome from the alignment data. Here, the example has unique paths.

novoBreak), and InfoGenomeR outperformed the other graph SV caller, JaBbA. Although CONSERTING and Weaver used discordant/split reads and read depths together, they exhibited a lower performance compared to DELLY2 and Manta. On the other hand, InfoGenomeR showed higher precision, even while maintaining the recall rate from the initial SV calls (Supplementary Fig. 3). In addition, InfoGenomeR remapped non-properly paired reads to unbalanced nodes to discover SVs at the intermediate step, which resulted in a 2.8% improvement in the recall rate for somatic SVs (Supplementary Fig. 4). As read-depth integration with SVs could be sensitive to variant size, we next compared the performances based on the variant size. Again, InfoGenomeR remained robust, regardless of variant size, and showed the highest precision and recall rate compared to all other tested methods (Supplementary Fig. 5). Finally, we compared InfoGenomeR, Weaver and JaBbA in terms of SVCNs detection, and yet again, InfoGenomeR showed better performance (Fig. 2).

For CNA breakpoint calling, InfoGenomeR exhibited an enhanced performance over BIC-seq2, likely due to its local segmentation strategy (Fig. 2). Specifically, InfoGenomeR predetermined CNA breakpoints using initial SVs (first-round iterations), discovered CNA breakpoints where candidate SVs existed (the intermediate step), and reduced false breakpoints in segmented regions by increasing the segmentation parameter (second-round iterations) (Supplementary Fig. 4). The local segmentation resolved trade-offs between filtering noises and recalling true variants, thereby resulting in the highest precision and recall. Although CONSERTING uses a similar local segmentation approach as InfoGenomeR, it showed a lower performance. Weaver, on the other hand, showed the lowest performance and was sensitive to variant size and purity (Supplementary Figs. 3, 5). JaBbA showed the second-best performance for CNA breakpoints. Next, we compared their performance in detecting integer CNs and ASCNs of segmented

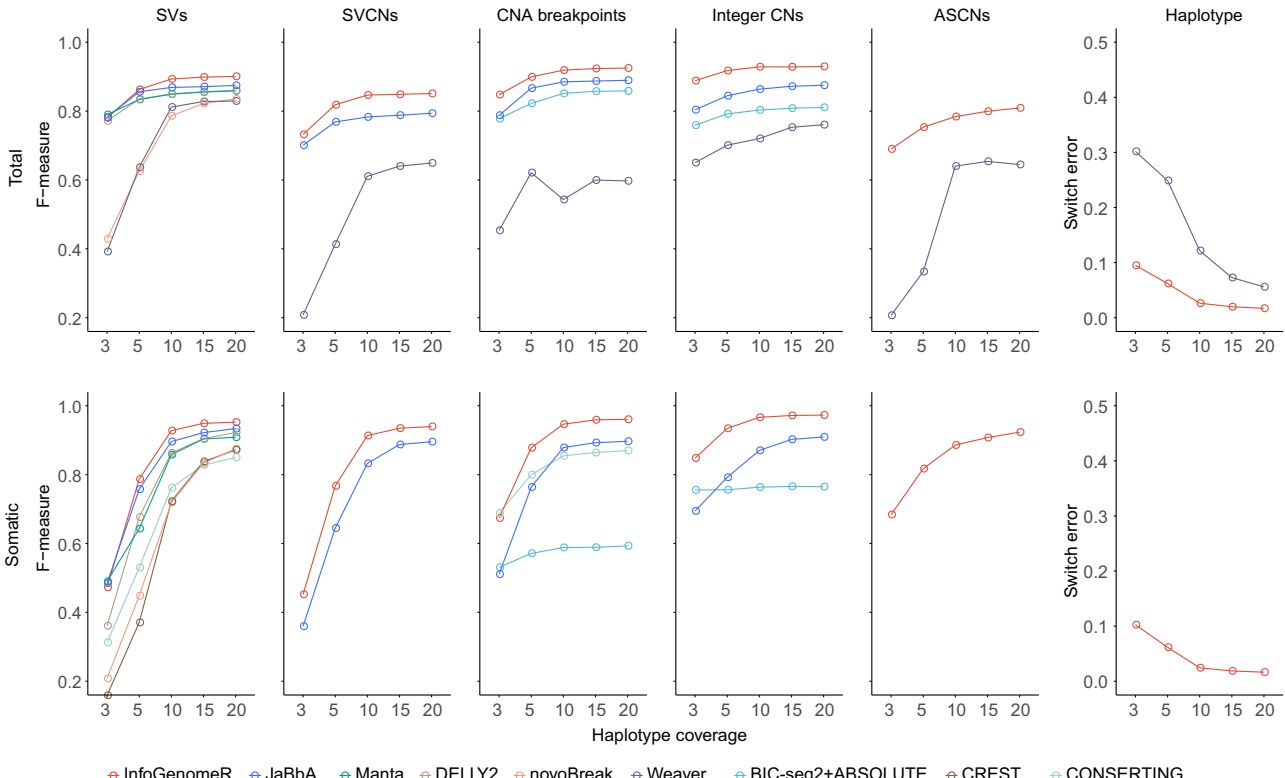

**Fig. 2 InfoGenomeR performance with simulated data sets.** F-measures were compared among variant-calling tools for five variant-calling categories (SVs, SVCNs, CNA breakpoints, integer CNs and ASCNs) and switch error rates for haplotype, with controls (somatic variants), and without controls (total variants including germline and somatic variants). The *x*-axis denotes the haplotype coverage (3× to 20×), representing the mean number of reads aligned to a nucleotide in a haplotype.

regions based on corroboration (defined as >90% of the regions of the segment having the same state, with true copy numbers)[10]. For integer CNs, Weaver was able to detect total integer CNs, but not somatic integer CNs with the germline coverage control. Further, we combined BIC-seq2 and ABSOLUTE to compare the ability to detect total and somatic integer CNs. JaBbA was compared with InfoGenomeR for both total and somatic integer CNs. InfoGenomeR showed an enhanced performance over the combination of BIC-seq2 and ABSOLUTE and outperformed JaBbA, achieving the best performance in detecting integer CNs. For ASCNs, InfoGenomeR showed an F-measure of 0.799 (15X), which was 14% higher than the F-measure of Weaver. For somatic ASCNs, InfoGenomeR showed the F-measure of 0.907 (15X). Since ASCNs depend on the number of SNPs, we observed that small germline variants (<10 kb) caused a bottleneck (Supplementary Fig. 5). InfoGenomeR showed F-measures of 0.940 (total variants) and 0.925 (somatic variants) for large ASCNs (>100 kb) (Supplementary Fig. 5).

For haplotype estimation, we measured the switch error rate between the true and inferred haplotypes, based on the total or somatic breakpoint graph. InfoGenomeR showed error rates of 1.98% and 1.87% for the total and somatic mode respectively (15X) (Fig. 2), and the small decrease in the error rate for the somatic mode might have resulted from the higher accuracy of somatic ASCN estimation. InfoGenomeR showed better performance for haplotype estimation than Weaver, because it could benefit from the better ASCN estimation than Weaver.

To compare the performance depending on the human reference genome versions, we evaluated InfoGenomeR against the five other tools using GRCh38-based simulated data sets. Performance gaps between SV callers were reduced compared with those of the GRCh37-based simulated data sets

(Supplementary Fig. 6). This reduction in performance gaps might have resulted from the mappability improvement of GRCh38. InfoGenomeR and JaBbA for total SVs, and InfoGenomeR and Manta for somatic SVs exhibited the best performances in that order, respectively. InfoGenomeR and JaBbA had similar performances for total CNA breakpoint calling. Although, the mappability improvement in the GRCh38 reference could reduce performance gaps among the variant-calling methods, the high performance of InfoGenomeR was still valid for the GRCh38 reference. Considering these results, InfoGenomeR outperformed the other variant-calling methods in all restricted variant-calling categories, for both the GRCh37 and GRCh38 references.

**Validation using cancer cell lines.** To evaluate the performance of InfoGenomeR, we analysed WGS data from three lung cancer cell lines (H292, A549 and H226)[29] and the HeLa cell line[30], whose karyotypes are well known. We constructed haplotype graphs for each cell line (Fig. 3). Because the graphs included multiple karyotypic possibilities as per the alternative Eulerian paths, we selected the one with the minimum entropy for the validation of karyotyping among the candidate karyotypes. The reconstructed karyotypes were matched with m-FISH karyotypes, and chromosomal ends predicted by InfoGenomeR were compared (Table 1, Supplementary Table 3 and Supplementary Fig. 7). InfoGenomeR identified 62.5, 50.0, 53.3 and 40% of the interchromosomal translocations from m-FISH (Table 1). Most of the unidentified translocations were found in centromeric or telomeric regions (Supplementary Table 3). For the correctly identified interchromosomal translocations, InfoGenomeR can detect breakpoints at the base-pair resolution in the haplotype, and types of complex SVs, such as chromothripsis that cannot be revealed by m-FISH.

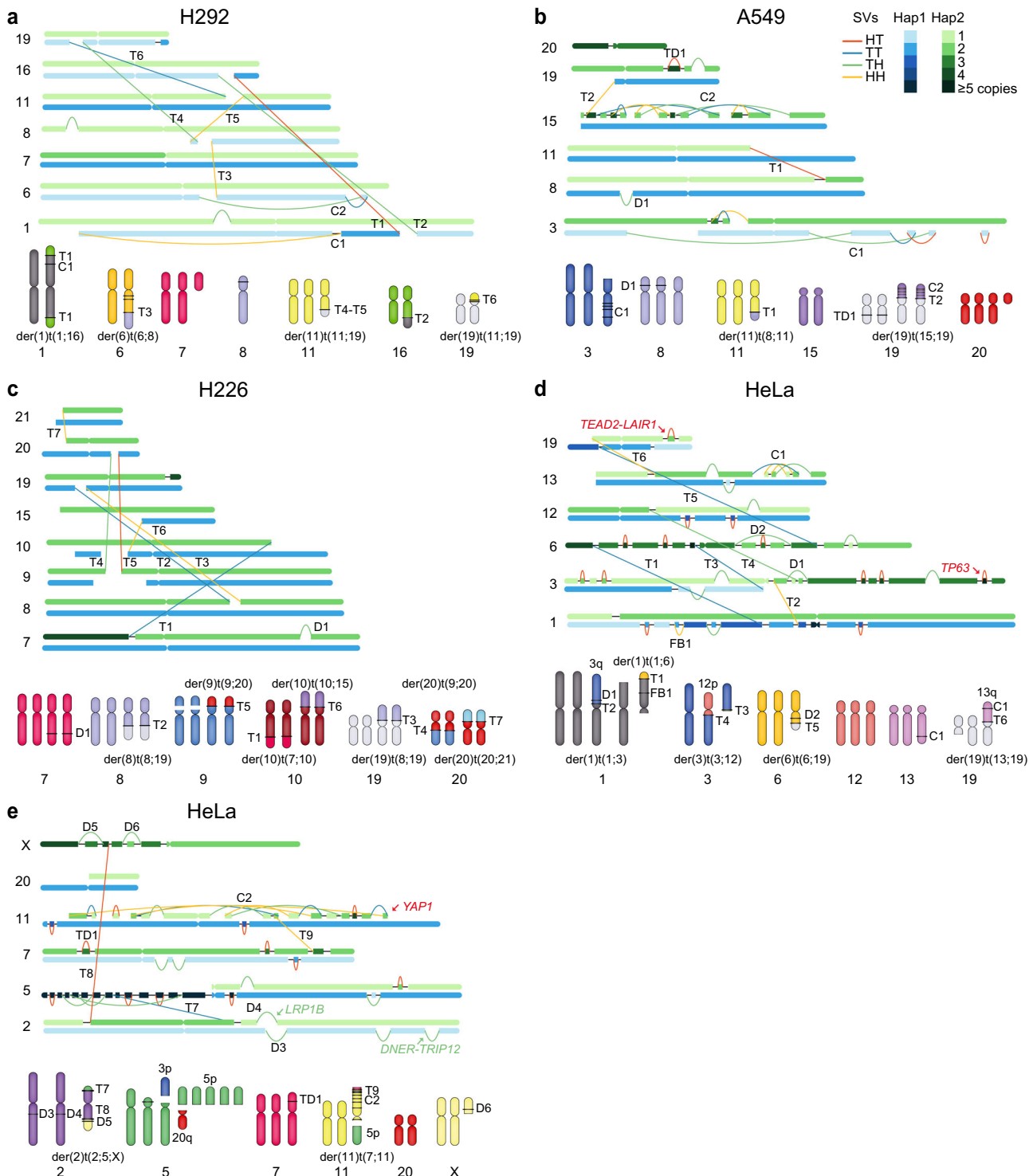

**Fig. 3 The haplotype graphs and reconstructed karyotypes of cancer cell lines.** The haplotype graph is composed of nodes and segment edges from two haplotypes (green and blue boxes) and reference edges and SV edges (black and coloured lines, respectively). The copy numbers of allelic segments are represented by the colour intensities (one to five copies). SVs included in karyotype analysis are denoted by D (deletion), TD (tandem duplication), T (translocation), FB (fold-back inversion) and C (complex SVs). **a–e** Sets of interchromosomal SVs across chromosomes and their karyotypes of the H292 (**a**), A549 (**b**), H226 (**c**) and HeLa (**d**, **e**) cell lines are shown.

The H292 cell line showed chromoplexy (rearrangement chains)[31] among chromosomes 6, 8, 11 and 19 (T3–T6 and C2 in Fig. 3a), resulting in der(6)t(6;8), der(11)t(11;19) and der(19)t(11;19). The A549 cell line was triploid and showed chromothripsis in chromosomes 3 and 15 (C1 and C2, respectively, in Fig. 3b). We reconstructed der(19)t(15;19)x2 that was generated

from chromothripsis of chromosome 15. In addition, we reconstructed the karyotype of the H226 cell line, which was tetraploid, with balanced translocations, t(8;19) and t(9;20) (T2-3 and T4-5, respectively, in Fig. 3c), and unbalanced translocations, t(7;10), t(10;15) and t(20;21) (T1, T6 and T7, respectively, in Fig. 3c). The derivative chromosomes were duplicated, which

**Table 1 Performance of karyotype reconstruction for cancer cell lines.**

| Cell line | Ploidy | Translocation | | Karyotype | |
|---|---|---|---|---|---|
| | | Precision | Recall | Precision | Recall |
| H292 | Diploidy | 1.000 (5/5) | 0.625 (5/8) | 0.800 (40/50) | 0.870 (40/46) |
| A549 | Triploidy | 1.000 (2/2) | 0.500 (2/4) | 0.924 (61/66) | 0.968 (61/63) |
| H226 | Tetraploidy | 0.875 (7/8) | 0.583 (7/12) | 0.775 (69/89) | 0.863 (69/80) |
| HeLa | Triploidy | 0.890 (8/9) | 0.400 (8/20) | 0.680 (53/78) | 0.828 (53/64) |

suggested that translocations were followed by whole-genome duplications (WGDs).

For the HeLa cell line (Supplementary Note 2 and Supplementary Fig. 8), we identified nine translocations, of which, eight matched the translocations identified by m-FISH. The unmatched translocation was between 3p and a near-centromeric region of the chromosome, representing centromeric noise (T3 in Fig. 3d). Notably, we reconstructed representative HeLa derivative chromosomes [der(1)t(1;3), der(12)t(3;12) and der(19)t(13;19)][32] with InfoGenomeR at the base-pair resolution (Fig. 3d). Our results showed that chromosome 11 had the excessive SVs with the loss of heterozygosity (LOH), implicating that chromothripsis underlay der(11)t(7;11) (Fig. 3e).

Further, we found *TP63* and *MYC* tandem duplications with arm-level amplifications and focal *YAP1* amplification in der(11)t (7;11) in the HeLa cell line (Supplementary Fig. 9); these amplifications are recurrent in cervical cancer[33]. In addition, we analysed changes in expression levels according to SVs using matched RNA-seq data[30] (Methods). We detected a homozygous exonic deletion in the *LRP1B* tumour suppressor gene and four head-to-tail or tail-to-head gene fusions (*DNER-TRIP12*, *SLC12A3-NLRC5*, *KLHDC4-SLC7A5* and *TEAD2-LAIR1*). These data were validated using discordant reads of matched RNA-seq data[30] (Supplementary Fig. 9). The gene expression in derivative chromosomes was upregulated proportionally to the increased copy number, as confirmed by the reconstructed karyotypes (Supplementary Fig. 10). Taken together, the reconstructed genome was supported by the earlier published report in cervical cancer and RNA expression data.

**InfoGenomeR can characterise complex SVs and karyotypes in cancers.** Having shown that InfoGenomeR could construct karyotypes of cancer cells, we applied InfoGenomeR to different data sets of breast invasive carcinoma (BRCA, *n* = 90)[20], glioblastoma multiforme (GBM, *n* = 37)[21] and ovarian serous cystadenocarcinoma (OV, *n* = 47)[22] taken from TCGA. InfoGenomeR identified 223, 124 and 275 somatic SVs on average from BRCAs, GBMs and OVs data sets, respectively, of which >20% were complex SVs (Supplementary Fig. 11). We performed clustering analysis of these complex SVs from the haplotype graph, defining an SV cluster as a set of closely rearranged focal segments (Supplementary Note 3 and Supplementary Fig. 12). The landscapes of somatic SVs and SV clusters of the total 174 data sets are described in the Supplementary Note 4. We further classified SV clusters from BRCA, GBM, and OV into three amplification types: (1) HSR (an SV cluster with high amplification (>10 copies) connected to a chromosomal arm); (2) HSR/DM (an SV cluster with high amplification connected to a chromosomal arm and a cycle with at least five multiplicities); and (3) DM (an SV cluster with a cycle with at least five multiplicities, without any connection to a chromosomal arm). HSR/DM amplification type represents either SV clusters of unclear distinction between HSRs and DMs or simultaneous existence of HSRs and DMs[12,34]. We also classified a deletion type chromothripsis, an SV cluster with interspersed LOH[19,35].

Next, we individually examined data from each cancer type. In the BRCA data set, we derived the karyotype structures of nine patients with rearrangement in chromosome 17 (Fig. 4a). In our results, chromosomes 11 and 17 were the most commonly rearranged chromosomes and exhibited HSR or HSR/DM type SV clusters. Also, HSRs and HSR/DMs accompanied CTs to generate derivative chromosomes. Our results showed that interchromosomal SVs caused frequent clustering of *ERBB2* on chromosome 17 with other amplified oncogenes in HSRs and HSR/DMs (Fig. 4a and Supplementary Fig. 13). Further, *CCND1* on chromosome 11 clustered most commonly with *ERRB2*, followed by *MECOM*, *FGFR1* and *MYC*. Taken together, these findings provide karyotypic evidence for the co-localisation of oncogenes and suggest that CTs are associated with the HSR, and HSR/DM frequently observed in BRCAs.

In the GBM data set, DMs, the main hallmarks of oncogene amplifications were observed in 16.2% (6/37) of the samples[4,36] (Fig. 4b). DMs were absent from chromosomes with excisions[34,37] or CTs, while the remaining segments were joined together to generate LOHs. We observed HSR/DMs in 59.4% (22/37) of the samples. Since DMs required a stringent condition of no connection to a chromosomal arm, most of the SV clusters were classified as HSR/DMs. The major GBM oncogenes, namely *CDK*, *MDM2*, *KIT*, *PDGFRA* and *EGFR*, were amplified in HSR/DMs and DMs (Supplementary Fig. 14). Further, *CDK4* and *MDM2* were the most frequently clustered partners on chromosome 12, while *KIT*, *PDGFRA* and *EGFR* showed interchromosomal clustering with *CDK4* and *MDM2*. Notably, their amplifications were high and focal, suggesting the possibility of DMs, which appeared to have developed via a mechanism different from that of the HSR observed in BRCAs.

OVs were characterised by arm-level CNAs and clusters of fold-back inversions suggesting breakage-fusion-bridge (BFB) cycles[11,38], which were common on chromosome 19 (*n* = 7, 14.9% of OVs). Fold-back inversions induce inverted repeats generating HSRs, and they are strongly associated with poor prognosis in OVs[38]. We observed HSRs with BFB cycles (fold-back inversions ≥5) in derivative chromosomes with interchromosomal SVs, where *BRD4* and *CCNE1* were frequently amplified on chromosome 19 (Supplementary Fig. 15). Along with BFB cycles, HSRs with CTs, similar to those in BRCAs, were also observed in derivative chromosomes with *BRD4* and *CCNE1* amplification, suggesting that different mechanisms could be involved in their amplifications.

**Application of multi-sample WGS data to reveal tumour evolution.** Tumour evolution has been investigated in the context of single nucleotide variants (SNVs) and CNAs[23,39,40]. However, differentiation between private and shared SVs in primary or metastatic cells have been less-thoroughly investigated due to false-positive SVs and inconsistent SV calling rates between multi-samples[41]. We applied InfoGenomeR to multi-sample WGS data from locally relapsed or metastatic breast cancer samples (Methods), downloaded from the EGA (EGAD00001002696)[23]. We

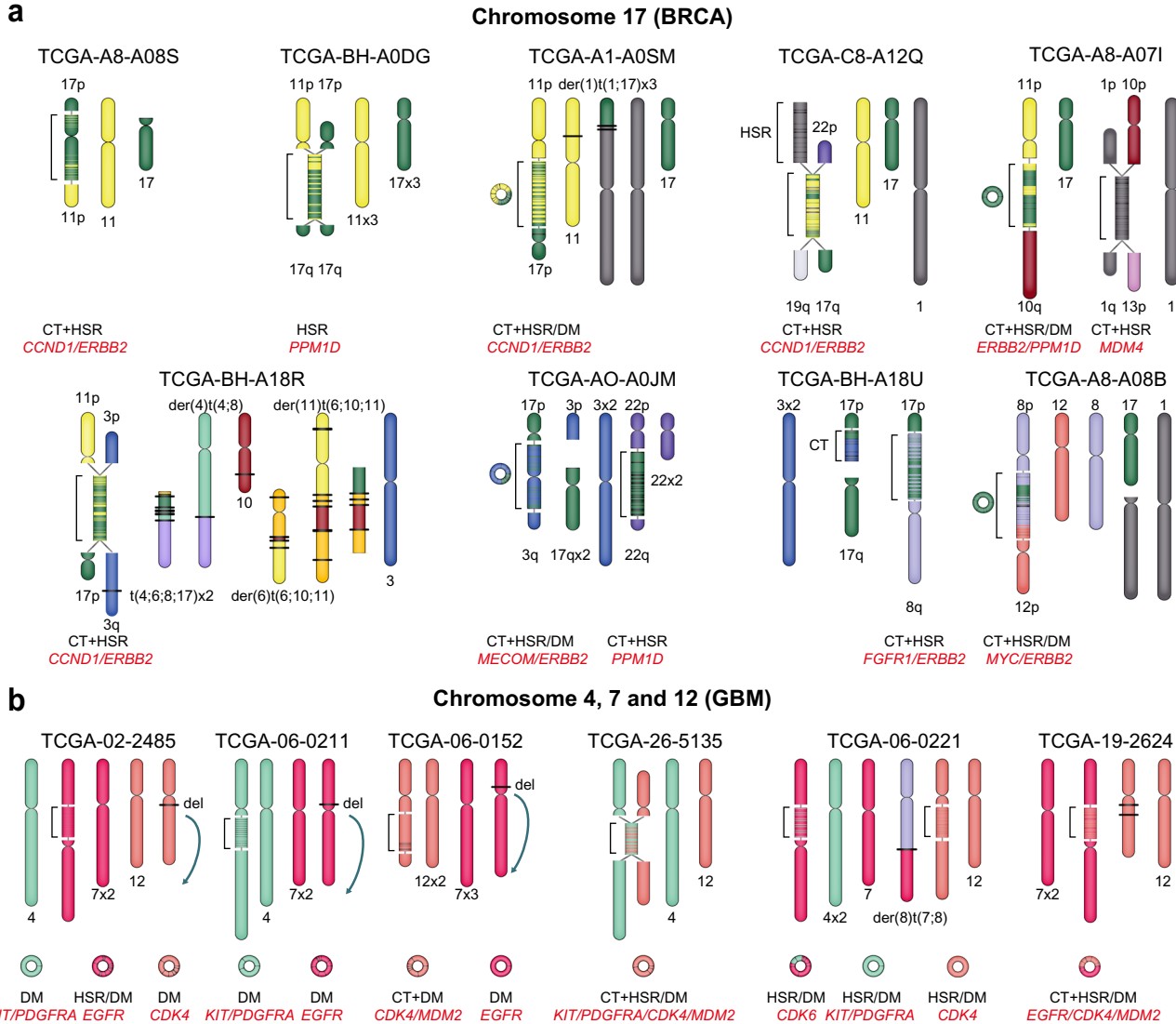

**Fig. 4 The karyotypic landscape of BRCAs and GBMs.** Karyotypic possibilities of BRCAs and GBMs from The Cancer Genome Atlas (TCGA), where SV clusters from the haplotype graph are represented in brackets. SV clusters are denoted by HSR, HSR/DM, DM and CT with concomitantly amplified cancer-related genes (red texts). The patient identifier of TCGA is shown at the top of each karyotype. Repetitive cycles that imply DM formations are represented as circles. **a** Commonly rearranged chromosomes 17 with interchromosomal SV clusters in BRCAs. SV clusters in BRCAs are connected to chromosomal arms or telomeric ends, forming derivative chromosomes with HSRs and CTs usually accompany them. Each chromosomal end at which SV clusters are located is represented with edges (grey lines). **b** Commonly rearranged chromosomes 4, 7 and 12, and DM formations in GBMs. DMs below chromosomal karyotypes are shown with amplified cancer-related genes in the SV clusters.

analysed 34 tumour samples from 15 patients with lesions described as primary and/or metastatic.

Six patients showed a higher accumulation of private SVs in primary tumours than in metastatic tumours (Fig. 5a). Two of these patients (PD4252 and PD4820) showed new SV clusters in primary tumours (Fig. 5b, c, Supplementary Fig. 16). Patient PD4252 had a LOH deletion of 9q in the primary tumour, where the remaining segments were incorporated into chromosome 1 with a LOH of 1p, forming an HSR with *NFIL3* amplification (PD4252a). Patient PD4820 had an HSR/DM with *ERBB2* and *BRD4* amplification and an HSR with *PAK1* amplification, and these were passed on during lymph node (LN) metastasis (PD4820c). A new SV cluster (cluster 3) identified in the primary tumour was generated with inverted repeats showing *FOXO3* amplification. These results indicate the acquisition of the SV cluster in primary tumours after LN metastasis. Although there is a possibility that a minor subclone without the SV cluster might have metastasised to the LN, we

discounted the idea since no sub-clonal CNA was observed in the primary tumours (Supplementary Fig. 16).

The results from the other nine patients showed that metastases were enriched in SV clusters that had either accumulated or were newly generated through evolutionary processes, thereby indicating metastatic evolution (Fig. 5a). Two of these patients (PD11460 and PD9193) showed new SV clusters in metastatic tumours. We found divergent evolution between the metastatic lesions and the primary tumour in patient PD11460. Further, the loss of 11p evolved only in the metastatic LN tumour (PD11460c), while a new cluster (PD11460 cluster 2) was generated in the metastatic skin tumour between chromosomes 8 and 11. This new cluster exhibited focal amplification of *FGFR1* and *CD82* (>10 copies), thereby developing an HSR in the derivative chromosome (Fig. 5d and Supplementary Fig. 16c). In patient PD9193, the primary tumour (PD9193a) had an SV cluster (PD9193 cluster 1) that was inherited by the metastatic LN

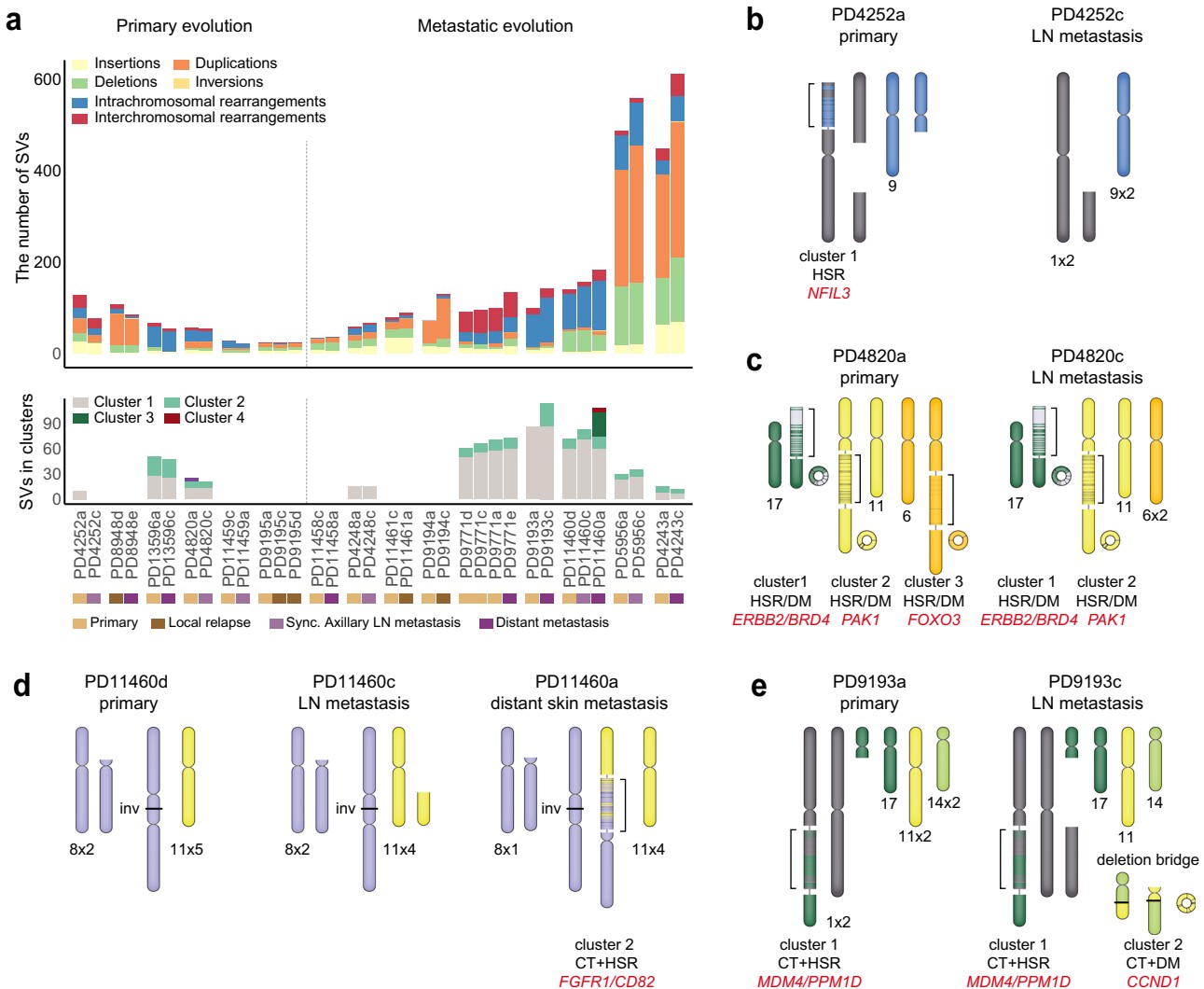

**Fig. 5 The evolution of breast cancer genomes. a** Bar plots of SVs and SV clusters discovered in metastatic and relapsed breast cancer from the European Genome–phenome Archive (EGA) data set[23]. The four cancer types studied included: primary tumour, local relapse, synchronous axillary LN metastasis and distant metastasis. The cancer types were sorted in the listed order, and the patients were classified as having primary or metastatic evolution depending on the accumulative patterns of SVs. **b** Patient PD4252 showed karyotypic evolution of chromosomes 1 and 9. **c** Patient PD4820 showed karyotypic evolution of chromosome 6. **d** Patient PD11460 showed karyotypic evolution of chromosomes 8 and 11. **e** Patient PD9193 showed karyotypic evolution of chromosomes 11 and 14. **b–e** The patient identifier of the EGA is shown at the top of each karyotype. Cancer-related genes are shown in red texts.

tumour (PD9193c) (Fig. 5e). A new DM (cluster 2) was generated by escaping from chromosome 11 with CT, encompassing focally amplified *CCND1* (>30 copies). The remaining segments of chromosome 11 had translocated to chromosome 14 by a small deletion bridge[31]. These results demonstrate the evolutionary processes of HSR and DM generation with CTs.

## Discussion

Our graph-based framework, InfoGenomeR, integrates individual variant callings for SVs and CNAs, purity and ploidy measurements, and haplotype estimation. Based on the breakpoint graph, InfoGenomeR establishes a haplotype graph, thereby narrowing down the target genome according to allele- and haplotype-specific information. As a result, it increases the scope of individual variant-calling and facilitates the identification of genome-wide SVs, thereby characterising the karyotypes of target genomes.

InfoGenomeR allows identification of complex rearrangement topologies (HSR, DM, HSR/DM and CT) in the reconstructed

cancer genome karyotypes. In a previous study, the identification of DM has been conducted using integrating SVs and CNAs[12], but the analysis was restricted to local amplified regions without recovering haplotype karyotypes. ShatterSeek[19] used an integrative approach of SVs and CNAs to identify CT; however, it did not provide karyotype structures such as derivative chromosomes and DMs resulted from CT. Recently, a decomposition method for DMs and/or linear chromosomes based on a haplotype graph has been introduced[42]. Nevertheless, this method lacks interpretation of other topologies such as HSRs, HSR/DMs or CTs. JaBbA introduced other complex topologies with DMs, except for karyotypes that were not derived from reconstructed haplotypes. InfoGenomeR enables us understand complex topologies with karyotype reconstruction simultaneously at the genome-wide level, as shown in the analysis of TCGA (Fig. 4) and EGA data (Fig. 5). InfoGenomeR can help identify the recurrent derivative chromosomes generated from chromosomes 11 and 17 with HSRs in BRCA. Our analysis of the SV clusters showed that *CCND1* and *ERBB2* were often closely clustered in these

derivative chromosomes. Besides, we found that GBMs and OVs were mainly characterised by HSR/DM or DM and HSR by fold-back inversions on different chromosomes.

CTs were recently reported to be found in more than half of cancers by ShatterSeek, where CTs with other complex events were more prevalent than canonical CTs that showed an oscillating pattern between two CN states[19]. However, the goal of ShatterSeek was restricted to figure out SV clusters of CTs, and the structures of derivative chromosomes were not investigated comprehensively because of the lack of a reconstruction strategy. Our results showed HSR, HSR/DM, or DM topologies involved with CT in chromosomal structures by reconstructing cancer genome karyotypes. We found that chromosome 17 is a template chromosome, which was recurrently rearranged with other chromosomes with CTs in BRCAs, demonstrating that complex events with CTs in multiple chromosomes generated derivative chromosomes. It was suggested that complex events involved with CTs in the formation of derivative chromosomes contributed to the amplification of cancer-related genes, such as CCND1, CDK4, MDM2 and ERBB2[19]. Our results showed that cancer-related genes were amplified in the formation of derivative chromosomes. Overall, we provided insights into the karyotypic view of complex rearrangements involved with CTs.

Through a multi-sample analysis, we could identify the evolutionary processes of HSR and DM generation with CTs, during metastatic tumour evolution. Previously, SVs have been investigated to emerge in metastasis. However, the discrimination between private and shared SVs was unclear, and karyotypic characterisation has not been performed. We observed that SVs could be misidentified as private SVs by a simple SV calling approach, even though they were shared SVs with supporting CNA evidence existing in primary and metastatic tumour. We performed imputation for candidate shared SVs that may exist in both primary and metastatic tumour during breakpoint graph construction, thus clearly distinguished true private SVs. These private SVs were shown together in HSR and DM topology with CT (Fig. 5d, e). We characterised their karyotypes by reconstructing derivative chromosomes and DMs, thus providing a basis for structure-based analyses of tumour evolution. Nevertheless, there were limitations in the current analysis. First, our applications to the primary and metastatic tumours were independent with each other even though we adjusted breakpoint graphs in the intermediate steps during iterative optimisations. In addition, we did not perform clone-specific interpretation, although sub-clonal SVs or CNAs may clarify tumour evolution processes. A joint approach for subclones across multi-samples[43] will be required for future analyses.

Despite the successful application of InfoGenomeR for genome-wide karyotype construction, there are clear opportunities to improve and extend this framework in the future. For instance, we filtered out potential false SVs at each iteration of the optimisation procedure, and they have not added to the graph, except at the intermediate SV-adding step (Fig. 1b). However, this may prevent recalls of true SVs. Furthermore, some translocations were not recalled from initial SV callings in the HeLa cell line, which might have occurred in centromeric or non-mappable repetitive regions. Long-read sequencing is required to find SVs that cannot be identified by short-read sequencing. Our framework is likely to benefit from long-read sequencing technologies so that SV calls from short-read sets could be integrated. Further, as the current framework constructs a representative graph for dominant tumour cells from the bulk WGS data, we eliminated minor changes in CN during the optimisation procedures. However, a few minor sub-clonal populations might have been removed during this process. Moreover, this elimination might be problematic for samples where sub-clonal populations form a considerable population (here, we

focused on nonclonal samples with a cancer purity >70%). The integration of deconvolution methods[17,18,43–45] for CNAs and SVs into our framework will further allow the investigation of sub-clonal structures and produce multiple breakpoint graphs for them. Finally, we have initiated the graph-based reconstruction of multi-sample genomes, thereby providing a basis for structure-based analyses. We propose that a phylogenetic method is now required to investigate karyotypic evolution (i.e. the measurement of an edit distance in the breakpoint graph). InfoGenomeR is not limited to cancers but can be used for other genetic diseases. A potential application is the analysis of somatic mutations in neurological diseases in which somatic SNVs reportedly contribute to the genetic diversity of human neurons[46]. Somatic SVs have not been investigated comprehensively in neurological diseases, although microscopic abnormalities have been shown[46]. The application of InfoGenomeR may discover genetic variations that have not been detected in SNVs.

In summary, we developed a method to reconstruct cancer genome karyotypes and explored the karyotypes of complex SVs in three cancer types (BRCAs, GBMs and OVs) and multi-sample data with primary and metastatic cancer cells. More cancer types should be explored to determine the wider karyotypic changes that occur during cancer development and evolution. We expect that cancer driver genes in these complex SVs can be used for identifying candidates for clinical treatment.

## Methods

**Initial SV detection by InfoGenomeR.** The variant callers DELLY2[6], Manta[7] and novoBreak[8] were used with default parameters to detect initial SVs with or without controls (total or somatic). Low-quality SVs, defined as <3 variant supporting reads or a mapping quality <20, were filtered out. Breakpoints of an SV were sorted by the chromosomal and coordinate order in the reference sequence, and the SV was annotated as head-to-head (HH), head-to-tail (HT), tail-to-head (TH) or tail-to-tail (TT) depending on the orientation of breakpoint adjacencies to the genomic segments. The head and tail are the 5′ and 3′ coordinates in the reference genome, respectively. Detailed settings for SV detection were shown in Supplementary Table 4. SV sets from individual SV callers were unified as the input of InfoGenomeR. Breakpoints predicted by the SV callers could differ for the same SV (if breakpoints of SVs were overlapped in <100 bp, they were considered as the same SV), and we empirically selected one of their breakpoints when SV sets were unified.

**Breakpoint graph construction.** InfoGenomeR constructs a breakpoint multi-graph $G(S, E)$ from genomic segments and SVs. A node set $S$ has two types, head nodes ($S_h$) and tail nodes ($S_t$), representing the head and tail sides of genomic segments, respectively. In the breakpoint graph, the $i$th genome segment is represented by a pair of the head and tail node, ($s_h^i$, $s_t^i$). An edge set ($E$) has three types: segment edge ($E_s$), reference edge ($E_r$) and SV edge ($E_v$). The segment edge connects the head node ($s_h^i$) and tail node ($s_t^i$) of the $i$th genomic segment, and the multiplicity of the segment edge represents the CN of the genomic segment. The reference edge connects the tail node ($s_t^i$) and the head node ($s_h^{i+1}$) between $i$th and $i + 1$th segment, representing the adjacency between adjacent genomic segments present in the reference genome. Conversely, the SV edge represents a novel adjacency between genomic segments that does not exist in the reference genome. The following iterative procedures are used to construct the breakpoint graph:

### Iterative step 1

*Local CN segmentation.* InfoGenomeR divided the genomic regions using current SV breakpoints. Then, in the pre-divided regions, it performed local CN segmentation with BIC-seq2[24] with the main penalty parameter $\lambda$, and measured the copy ratio between observed and expected read counts (from a control, if available) in the genomic regions. Briefly, BIC-seq2 uses the Bayesian information criterion to determine breakpoints that are composed of two terms[24]: the negative log likelihood term, which explains how well the model with the breakpoint fits the read-depth data, and the penalty term, which is proportional to the number of breakpoints and prevents over-segmentation. The parameter $\lambda$ adjusts the penalty term, with higher $\lambda$ preventing excessive breakpoints. InfoGenomeR used different $\lambda$ for first- and second-round iterations, where the second iterations used a higher penalty, thereby allowing the mis-segmented regions without SV evidence to be merged. In the present analysis, the parameter values of bin size = 100, initial $\lambda = 1$ and final $\lambda = 16$ were used for the simulated data. Cancer cell line data showed a higher noise level than the simulated data, and the parameter values of bin size = 100, initial $\lambda = 1$ and final $\lambda = 2000$ were used for cancer cell line data, where the reconstructed karyotypes were well matched

with the m-FISH karyotypes. The same parameters used in cancer cell line data were used for TCGA and EGA data subsequently.

## Iterative step 2

*Purity and integer CN estimation.* The copy ratios of genomic segments were measured by local CN segmentation, and the cancer purity ($p$) and ploidy ($\tau$) were estimated using ABSOLUTE[25]. The end sides of a genomic segment were represented by a head and tail node, and the copy ratio and integer CN of the genomic segment of the node $s$ (head or tail) were denoted by copyratio($s$) and CN($s$), respectively. The copyratio($s$) was fitted with a Gaussian mixture model, each component of which was a Gaussian distribution representing the integer CN state ($q$) with a mean copy ratio, $m_q = \{qp + 2(1 - p)\}/D$. Here, $q$ took an integer CN (0, 1, 2,...) in the cancer genome, and $D = p\tau + 2(1 - p)$ was the average ploidy of cancer and normal cells. ABSOLUTE estimated cancer purity and ploidy from the copy ratios, and the integer copy number CN($s$) was assigned according to the highest posterior probability of integer copy number of the states from the Gaussian mixture model. Because ABSOLUTE assigns the predefined maximum integer CN when the copyratio($s$) is larger than the estimation limit, in this case, we calculated non-integer CN'($s$) satisfying the copy ratio equation, copy ratio($s$) = $\{CN'(s)p + 2(1 - p)\}/D$. Then, CN($s$) is assigned as [CN'($s$)], rounding of the non-integer CN'($s$). Perturbations were performed in low-confidence segments and their integer CNs were decided during the next integer programming step. The segment of the node $s$ was defined as low-confidence if: (1) the posterior probability of the integer CN, $p(CN(s)) < 0.95$, (2) the segment size($s$) < 50 bp or $s$ had no depth information (unmappable regions) or (3) |CN($s$) − CN'($s$)| > 0.35 for high CNs. The purity estimations were repeated during the InfoGenomeR iterations, and the final purity was decided in the last iteration.

## Iterative step 3

*Integer programming for finding edge multiplicities.* At this step, the optimal breakpoint graph representing the cancer genome was reconstructed, where the multiplicities of edges satisfied the CN balance condition (Eq. (1))[26]. The CN balance condition ensured Eulerian paths from one telomere to another for each linear chromosome. The multiplicity of an edge ($e$) was denoted by $\mu(e)$, and a segment edge, SV edges (multiple SVs can exist) and a reference edge adjacent to a node ($s$), were denoted by $e_s(s)$, $E_v(s)$, and $e_r(s)$, respectively. The multiplicity of a segment edge $\mu(e_s(s))$ is the sum of the multiplicity of a reference edge $\mu(e_r(s))$ and multiplicities of SV edges $\mu(e_v(s))$ for node $s$.

$$\mu(e_s(s)) = \mu(e_r(s)) + \sum_{v \in E_v(s)} \mu(v), \ \forall s \in \ S\backslash\text{telomeric ends} \quad (1)$$

The multiplicities of edges satisfying the CN balance condition were determined by integer programming. For a confident segment edge, the multiplicity was given by the integer CN of the segment, which was an estimate of the previous copy (constant if the segment of the node was confident). To determine multiplicities of variable edges (reference and SV edges, and low-confidence segment edges), we first found an interrelated subset of nodes ($S_\text{related}$) and then solved the integer programming problem (Eq. (2)) to find the multiplicities of edges adjacent to interrelated nodes but independent from the other subsets. The interrelated subset was defined inductively by including adjacent nodes, Adj($s$) from the start node ($s_\text{start}$) and could be found in a breadth-first search (BFS) manner. Note that if the BFS encountered a confident node, it stopped to propagate in the segment edge direction (the adjacent node by the segment edge, $\text{Adj}_s(s)$ was not included, whereas the other adjacent nodes by the reference and SV edges, $\text{Adj}_{r,v}(s)$, were included). For any $s_\text{start}$, $S_\text{related} = \{s_\text{start}\}$ was constructed, and then $S_\text{related}$ was expanded as follows.

$$Adj_{r,v,s}(s) \subset S_\text{related} \text{ if CN}(s) \text{ is low-confident, } \forall s \in S_\text{related}$$
$$Adj_{r,v}(s) \subset S_\text{related} \text{ if CN}(s) \text{ is confident, } \forall s \in S_\text{related}$$

Given the constant integer CN states of segments in each interrelated subset, the multiplicities of reference edges and SV edges were decided with a small perturbation of integer CNs of low-confidence segments. An optimisation problem was defined to satisfy the CN balance condition (Eq. (1)) for all nodes in $S_\text{related}$. The lpSolveAPI R package was used to solve the integer programming problem.

$$\text{Minimise} \sum_{s \in S_\text{related}} (\mu(e_s(s)) - \mu(e_r(s)) - \sum_{v \in E_v(s)} \mu(v))$$

subject to

$$\mu(e_s(s)) \geq \mu(e_r(s)) + \sum_{v \in E_v(s)} \mu(v)$$
$$\sum_{v \in E_v(s)} \mu(v) \leq |\mu(e_s(s)) - \mu(e_s(\text{Adj}_r(s)))| \quad (2)$$

If size($s$) < 50 bp, $\mu(e_s(s)) \in [0, \text{max CN}]$,
else if $s$ is low-confident, $\mu(e_s(s)) \in \{CN(s), \text{alternative CN}(s)\}$,
else, $\mu(e_s(s)) = CN(s)$.

The first constraint prevented nonsense solutions wherever adjacencies exceeded CNs of segments. The second constraint was for an upper bound of the multiplicities of SV edges, which did not exceed the difference between multiplicities of adjacent segment edges in the SV breakpoint. This maximally preserved the existing reference edges

between adjacent segment edges. For rare cases where SV breakpoints were exactly reciprocal, SVs could be filtered out by the second constraint, and to restore them, a virtual (zero-length) segment was left between the reciprocal breakpoints. The third to fifth constraints were for integer CNs of segments. If the size of the segment was too small to measure the CN or if mis-segmentation by SV breakpoint errors occurred, the CN was imputed between zero and the maximum CN. Here, the minimum size threshold was set at 50 bp. For the segments >50 bp, if $s$ was confident, the multiplicity was fixed by the original estimate CN($s$); otherwise, the multiplicity changed within an alternative-CN range, alternative CN($s$), which was set to the next best integer state from ABSOLUTE in the current analysis.

In cases with multiple solutions, the one with (1) the maximum multiplicities of SV edges and (2) the multiplicities of segment edges closest to the initial CNs is selected. Maximising SV edges recalls true SVs as much as possible while it still excludes false SVs (false SVs hardly satisfy the CN balance condition), and prevents null solutions in cases where SV multiplicities become zero, such as simple inversions and balanced translocations. The solution can be found by gradually changing the bounding constraints for SV and segment edges.

$$\sum_{s \in S_\text{related}} \mu(e_v(s)) > \text{the mininum bound for SV edges}$$

$$\sum_{s \in S_\text{related}} |\mu(e_s(s)) - CN(s)| < \text{the maximum bound for segment edges}$$

Notably, SVs with zero multiplicities are false positives and are removed before the next iteration.

The iterative steps restart with the SV set obtained after filtering out SVs with zero-multiplicity. The different $\lambda$ parameter values in BIC-seq2 are used for the first and second iterations. Before the second iterations and settling the breakpoint graph, SV edges are added by remapping discordant and unmapped reads to de novo references[47] from candidate adjacencies (Supplementary Note 5). For the somatic mode (a control exists), germline variants are excluded, and additional processes are performed to reconstruct cancer genome graphs with somatic SVs (Supplementary Note 6).

In addition, after breakpoint construction, SVs are classified as simple or complex SVs, based on the respective breakpoint graph (Supplementary Note 7). Germline variants and short simple SVs (<100 kb) are bottlenecks for karyotype reconstruction, because they do not have sufficient allelic information for the allele-specific graph and may cause an over-segmentation of the genome. Assuming that they are negligible in the karyotyping view, we simplify the breakpoint graph by removing SV edges and CN bins for germline variants (Supplementary Note 8 and Supplementary Fig. 17).

**Allele-specific CN estimation.** In addition to integer CNs based on total read depths, read depths of heterozygous SNPs provide information about allele-specific CN. The integer CN, $\mu(e_s(s))$ of each segment from the breakpoint graph is divided into allele-specific CNs, ASCN($s$), using heterozygous SNPs (if a control exists, all the heterozygous SNPs in the control are used). Let $A = \{A_1, A_2, ..., A_{[(\mu(e_s(s)) + 1)/2]}\}$ denote all the possible states of allele-specific CNs that the genomic segment can have, where the integer CN can be divided into a set of $[(\mu(e_s(s)) + 1)/2]$ possible cases, and for each $A_i = \{A_{i,1}, A_{i,2}\}$, $A_{i,1} + A_{i,2} = \mu(e_s(s))$. For example, if the multiplicity of the segment edge, $\mu(e_s(s)) = 3$, there are two cases, $A_1 = \{0, 3\}$ and $A_2 = \{1, 2\}$. Given the $A_i$ of each segment, the read depths of $N$ heterozygous SNPs, $o_j = (o_{j,1}, o_{j,2})$, can be fitted using negative binomial (NB) distributions when the allele-specific copy numbers for each SNP, denoted by $a_j = (a_{j,1}, a_{j,2})$, are given. The pair of SNP depths, $o_{j,1}$ and $o_{j,2}$, are observed from $a_{j,1}$ and $a_{j,2}$, respectively. Here, the allele-specific copy numbers of heterozygous SNPs are latent variables.

$$p(O|\Theta) = \prod_{j=1}^{N} \sum_{a_j} p(o_j, a_j | b, p, \phi_1, \phi_2) \quad (3)$$

$$p(o_j|a_j, b, p, \phi_1, \phi_2) = \text{NB}(o_{j,1}|b(pa_{j,1} + (1 - p)), \phi_{j,1})\text{NB}(o_{j,2}|b(pa_{j,2} + (1 - p)), \phi_{j,2}) \quad (4)$$

Given the purity $p$ measured from the previous breakpoint graph construction, $p(O|\Theta)$ per segment is maximised by estimating the haplotype base coverage $b$ and dispersion parameters $\phi_1$ and $\phi_2$ of the negative binomial distributions for $A_{i,1}$ and $A_{i,2}$, respectively. The EM algorithm is used to estimate the maximum likelihood parameters for a given $A_i$ (Supplementary Note 9). The maximum likelihood, $\hat{L}(A_i)$ and the likelihood score, $\text{Score}_L(A_i)$ for each $A_i$ are obtained from the iterative divisions of $\mu(e_s(s))$, and ASCN($s$) = $\hat{A}$ such that maximised likelihood score is selected.

$$\hat{L}(A_i) = P(O|\hat{\Theta}_i) \quad (5)$$

$$\text{Score}_L(A_i) = \frac{\hat{L}(A_i)}{\sum_j \hat{L}(A_j)} \quad (6)$$

$$\hat{A} = \arg\max_{A_i} \hat{L}(A_i) \quad (7)$$

Nevertheless, not all of the initial estimations of ASCN($s$) were confident, and ASCN($s$) were defined as low-confidence if (1) $\text{Score}_L(\hat{A}) < 0.8$, or (2) the number of

heterozygous SNPs <5. For the low-confidence segments, we searched for the best ASCNs that minimised the objective function during the next round of allele-specific breakpoint graph construction.

**Allele-specific breakpoint graph construction.** Based on the ASCNs, an allele-specific breakpoint graph AG(S, E) was constructed, where the node set $S = \bar{S} \cup S_1 \cup S_2$ was composed of balanced ($\bar{S}$) and imbalanced nodes ($S_1$ and $S_2$ for temporal two haplotypes), which denote the heads and tails of genomic segments with balanced and imbalanced ASCNs, respectively. In the allele-specific breakpoint graph, the imbalanced nodes are assigned to haplotype 1 or haplotype 2 temporally, whereas the balanced nodes are not assigned. The phased states (haplotype 1 and haplotype 2) of imbalanced nodes are preserved within the imbalanced ASCNs and can be switched across genomic segments with balanced ASCNs.

In detail, genomic segments with imbalanced ASCNs, named imbalanced AS segments, were represented by two head ($S_{1,h}$ and $S_{2,h}$) and tail nodes ($S_{1,t}$ and $S_{2,t}$). Genomic segments with balanced ASCNs, named balanced AS segments, were represented in the same way as in the breakpoint graph. Thus, the multiplicities of the segment edges for the imbalanced and balanced segments were ASCNs and total copy numbers, respectively. The allele-specific graph implied that if the multiplicities of segment edges are imbalanced, the SV edges can be assigned to one of the alleles. In the case of imbalanced AS segments, differences between adjacent segments depended on the temporal phased state of AS segments, such that we could assign SV edges aligning imbalanced AS segments uniquely to satisfy the copy balance condition. However, with balanced AS segments, the phased state of AS segments and SV edges was not determined uniquely since the value of the objective function did not dependent on the phased state. The allele-specific breakpoint graph was constructed by following integer programming problem. To adjust multiplicities for low-confidence segments, all the candidate integer divisions were searched with a penalty function $\varepsilon$ added to the objective function. This was proportional to the rank of the likelihood score of the integer divisions for low-confidence segments, and was zero for the confident segments. Since the penalty of the low-confidence segment was added twice for the head node and tail node for the segment, the penalty was divided by two while adding it to the objective function.

$$\text{Minimise} \sum_{s \in S} (\mu(e_s(s)) - \mu(e_r(s)) - \sum_{v \in E_v(s)} \mu(v) + \varepsilon(s)/2)$$

subject to

$$\mu(e_s(s)) \geq \mu(e_r(s)) + \sum_{v \in E_v(s)} \mu(v)$$

$$\sum_{v \in E_v(s)} \mu(v) \leq |\mu(e_s(s)) - \mu(e_s(\text{Adj}_r(s)))|$$

If $s$ is low-confident,

$$\{\mu(e_s(s_1)), \mu(e_s(s_2))\} \in \{\text{ASCN}(\hat{s}), \text{alternative ASCN}(\hat{s})\} \text{ for } s \notin \bar{S},$$

$$\mu(e_s(s)) = \mu(e_s(\hat{s})) \text{ for } s \in \bar{S},$$

else,

$$\{\mu(e_s(s_1)), \mu(e_s(s_2))\} = \text{ASCN}(\hat{s}) \text{ for } s \notin \bar{S},$$

$$\mu(e_s(s)) = \mu(e_s(\hat{s})) \text{ for } s \in \bar{S}. \qquad (8)$$

This integer programming problem, where $s$ in the objective function can be $s_1$ and $s_2$ for imbalanced AS segments, requires exponential time for $\mu(e_s(s_1))$ and $\mu(e_s(s_2))$ to alternate between $\text{ASCN}_1(\hat{s})$ and $\text{ASCN}_2(\hat{s})$, or $\text{ASCN}_2(\hat{s})$ and $\text{ASCN}_1(\hat{s})$, respectively, when the ASCN measurement of the node $\hat{s}$ is denoted by $\text{ASCN}(\hat{s}) = \{\text{ASCN}_1(\hat{s}), \text{ASCN}_2(\hat{s})\}$. Here, $\hat{s}$ denotes a node in the previous breakpoint graph, which can be expanded to $s_1$ and $s_2$ or remain intact if $\text{ASCN}(\hat{s})$ is balanced. To solve the integer programming problem in the series of imbalanced AS segments, a heuristic was used to determine the multiplicities of segment edges in a greedy manner. The detailed objective function and the heuristic method are described in the Supplementary Note 10 and Supplementary Fig. 18.

**Haplotype segments.** Haplotype segments $H = \{H_1, H_2, ..., H_n\}$ are defined from the allele-specific graph AG(S, E), and each element $H_i = \{H_{i,1}, H_{i,2}\}$ is a set of imbalanced AS segments for haplotype 1 and haplotype 2, where heterozygous SNPs are phased by (1) allelic imbalances, and (2) focal (<1 Mb) nonhomologous SVs. First, sets of consecutive imbalanced AS segments between the balanced AS segments in the allele-specific graph are collected, where the phase of imbalanced AS segments in each set was determined by the integer programming in the previous allele-specific breakpoint graph construction section. Then, SVs between segments from two sets of imbalanced AS segments were classified into homologous (>100 bp homology) and nonhomologous SVs (≤100 bp homology)[48], and subsequently the presence of nonhomologous SVs was checked between the imbalanced AS segments (Supplementary Fig. 19a). The focal nonhomologous SVs were assumed to occur in a single allele, excluding rare possibilities in which homologous chromosomes were exchanged by nonhomologous mechanisms at the same focal breakpoints. The assumption simplified the haplotype phasing problem by preventing incorrect allelic switching (haplotype switching errors), and the sequences of imbalanced AS segments were merged into a haplotype segment. For

example, if two sequences of imbalanced segments, namely, $k, k + 1, ..., k + k'$ segments and the $l, l + 1, ..., l + l'$ segments, existed and nonhomologous SVs were found between them, then, the haplotype segment was defined as:

$$H_{i,1} = \{(h_{1,h}^k, h_{1,t}^k), ..., (h_{1,h}^{k+k'}, h_{1,t}^{k+k'}), (h_{1,h}^l, h_{1,t}^l), ..., (h_{1,h}^{l+l'}, h_{1,t}^{l+l'})\} \qquad (9)$$

$$H_{i,2} = \{(h_{2,h}^k, h_{2,t}^k), ..., (h_{2,h}^{k+k'}, h_{2,t}^{k+k'}), (h_{2,h}^l, h_{2,t}^l), ..., (h_{2,h}^{l+l'}, h_{2,t}^{l+l'})\} \qquad (10)$$

The head and tail node pair $(h_{1,h}^k, h_{1,t}^k)$ indicated the $k^{th}$ segment of haplotype 1, which was assigned based on the nodes from the allele-specific graph ($s_{1,h}^k, s_{1,t}^k$) or ($s_{2,h}^k, s_{2,t}^k$). The assignment was determined by the integer programming for the copy number balance condition in $H_i$, with the constraint for nonhomologous SVs (Supplementary Note 11). SNPs in the haplotype segment were phased by maximising the likelihood in Eq. (4), given the ordered state of imbalanced AS segments from the haplotype segment. For example, if $o_{j,1}$ was observed from $\text{snp}_{j,1}$ in the $k^{th}$ segment in $H_i$, the heterozygous SNPs, $\text{snp}_{j,1}$ and $\text{snp}_{j,2}$ would correspond to $H_{i,1}$ and $H_{i,2}$, respectively, if,

$$p((o_{j,1}, o_{j,2})|(\mu(e_s(h_{1,h}^k)), \mu(e_s(h_{2,h}^k))), \Theta_k) > p((o_{j,2}, o_{j,1})|(\mu(e_s(h_{1,h}^k)), \mu(e_s(h_{2,h}^k))), \Theta_k).$$

**Haplotype breakpoint graph construction.** Previously, in the allele-specific breakpoint graph, the sequences of imbalanced AS segments and nonhomologous SVs defined haplotype segments $H$, and heterozygous SNPs in imbalanced ASCNs of haplotype segments were phased. The haplotype breakpoint graph was constructed by phasing SNPs in balanced AS segments using population information and determining the end-to-end order of the haplotype segments. A haplotype was obtained by using a constrained version of the Viterbi algorithm for the hidden Markov model (HMM) of BEAGLE, where the transition and emission probabilities were defined from the localised haplotype-cluster graph[27]. As imbalanced heterozygous SNPs were already phased in haplotype segments, the Viterbi path was enforced to follow the phased order of SNPs (Supplementary Note 12 and Supplementary Fig. 19b, c). The Viterbi path decided the order of haplotype segments, while simultaneously phasing heterozygous SNPs in balanced ASCNs. Finally, the haplotype graph HG(S, E) was constructed. We denoted haplotype-specific copy numbers, which were derived from the haplotype phasing of allele-specific copy numbers of the node $\hat{s}$ (indicating $\hat{s}_1$ or $\hat{s}_2$) from the allele-specific graph, as $\text{HSCN}_1(\hat{s})$ and $\text{HSCN}_2(\hat{s})$. For the imbalanced nodes, $\text{HSCN}_1(\hat{s})$ and $\text{HSCN}_2(\hat{s})$ were $\mu(e_s(\hat{s}_1))$ and $\mu(e_s(\hat{s}_2))$ or $\mu(e_s(\hat{s}_2))$ and $\mu(e_s(\hat{s}_1))$, respectively, depending on the phased states of heterozygous SNPs in the imbalanced segments. For the balanced nodes, $\text{HSCN}_1(\hat{s}) = \text{HSCN}_2(\hat{s}) = \mu(e(\hat{s}))/2$. After the haplotype-specific copy numbers were obtained, a haplotype graph was constructed by following integer programming. In other words, the multiplicities of segment edges from imbalanced nodes were ordered with expansions of balanced nodes from the allele-specific graph, and the multiplicities of reference and variant edges were assigned to minimise the objective function.

$$\text{Minimise} \sum_{s \in S} (\mu(e_s(s)) - \mu(e_r(s)) - \sum_{v \in E_v(s)} \mu(v))$$

subject to

$$\mu(e_s(s)) \geq \mu(e_r(s)) + \sum_{v \in E_v(s)} \mu(v)$$

$$\sum_{v \in E_v(s)} \mu(v) \leq |\mu(e_s(s)) - \mu(e_s(\text{Adj}_r(s)))| \qquad (11)$$

$$\mu(e_s(s_1)) = \text{HSCN}_1(\hat{s})$$

$$\mu(e_s(s_2)) = \text{HSCN}_2(\hat{s})$$

**Enumeration of Eulerian paths.** To identify the candidate genomes, Eulerian paths were enumerated to alternate between segment edges and SV/reference edges on the haplotype graph constructed in the previous step. Head and tail nodes that did not satisfy the copy number balance condition (including original telomere ends) were considered as ends of the reconstructed chromosomes $P$, which could also be true ends or breaks due to missing SVs or miscalculated CNs. Circular chromosomes, $C$, included in the DM cluster were observed as circular paths. Eulerian decomposition problem (EDP) was defined to find linear and circular chromosomes from the breakpoint graph[42]. Although the min-EDP, which minimised the number of paths and cycles, $|P| + |C|$, was previously suggested to describe the most possible karyotype[42], the min-EDP was not always biologically relevant (i.e. the max-EDP could be the case). In this study, we formulated minimum-entropy Eulerian path enumeration that prioritised the decomposition of $P$ and $C$ with the minimum entropy. To enumerate candidate Eulerian paths, we used a multiway tree structure[28], in which each tree node represents the pairing state of the breakpoint graph edges. The multiway tree was expanded in a root-to-leaves model by sequentially increasing the level and processing of each node in the breakpoint graph (Supplementary Fig. 20 and Supplementary Note 13). Leaf nodes represented possible edge-pairing states delineating Eulerian paths reaching every genomic segment.

The enumeration for all the Eulerian paths is an NP-hard problem, and we prioritised Eulerian paths with the minimum entropy as biologically relevant cases. First, connected chromosomes were segregated, and the enumerations were performed inside connected chromosomes. For highly segmented genomes, the

haplotype breakpoint graph could be simplified by further excluding simple SVs (tandem duplications, deletions, and block-interchange insertions), which did not significantly affect candidate karyotypes. Then, a minimum-entropy search was applied to prioritise solutions with the minimum entropy. For instance, a node in the multiway tree at level $l$ represented edge-pairing states of a total of $l$ node in the breakpoint graph, and there were $n$ distinct paths. The total number of paths was $w = w_1 + w_2 + ... + w_n$, where $w_i$ is the multiplicity of $i$th path. The entropy at level $l$, $e_l$, was derived from the following formula:

$$e_l = -\sum_{i}^{n}(w_i/w)\log(w_i/w) \qquad (12)$$

The low entropy indicated that, a set of SVs was duplicated through additional amplification processes such as arm-level duplications, WGDs, and other duplication processes in HSRs and DMs. This required a shorter distance than individual occurrences of SVs. Branches from the multiway tree were cut if the solution space grew too rapidly, and candidate genomes were obtained from the leaf nodes of remaining branches.

**Breakpoint graph construction for multi-sample data**. For multi-samples, we unified SV sets from the initial SVs of each sample. Using the unified SV sets, the breakpoint graph of each sample was constructed. Then, we classified SVs into private and shared SVs depending on the existence of raw SV evidence (discordant or split reads) (Supplementary Table 5). If a shared SV is not called in a primary tumour, it is observed as a private SV in a metastatic tumour and vice versa. This approach using the unified SV sets has an advantage that uncalled SV edges in each sample can be added to the graph if there are supporting copy number depths and adjacent SV information in that sample. For the cases with difficulties in distinguishing private and shared SVs, we made another round of iterative optimisation (Supplementary Note 14).

**Simulated data set**. First, 12 simulated normal-tumour pairs were created from the NA12878, HG00732, NA19238 and HG00513 individuals in phase 3 of the 1000 Genomes Project. We simulated approximately 3000 germline and 200 somatic SVs per cancer genome, where the proportion and size of each SV type were derived from the previous studies[2,48] (Supplementary Note 1, Supplementary Table 6 and Supplementary Fig. 21). Diploid to tetraploid cancer genomes were generated by WGD operations, and each cancer genome was mixed with the matched normal genome with a different purity (60, 75, and 90%). We generated Illumina HiSeq 2 × 100 reads from heterogeneous genomes with 3X, 5X, 10X, 15X, and 20X haplotype fold coverages using ART (version 2.5.8)[49], and reads were mapped to the GRCh37 reference genome using Burrows-Wheeler Aligner-Maximal Exact Matches (BWA-MEM)[50]. To compare performance depending on the human reference genome versions, we generated diploid to tetraploid cancer genomes from NA12878 based on the GRCh38 reference using the same simulation schemes, and the reads were mapped to the GRCh38 reference genome.

**Data acquisition and preprocessing**. For data acquisition, the SRA Toolkit (version 2.8.2)[51] was used to download WGS and RNA-seq data of the HeLa cell line and WGS data of lung cancer cell lines. The GDC client (version 1.2.0)[52] was used to download WGS data of TCGA samples. The EGA client (version 2.2.2)[53] was used to download WGS data of metastatic breast cancers.

Reads from paired-end and mate-pair libraries of the HeLa genome were mapped to the human reference genome (GRCh37) using BWA-MEM[50] with default parameters (version 0.7.15). DELLY2, Manta, and novoBreak were used for SV callings from paired-end data, and DELLY2 was used for mate-pair data. Initial SV callings from both libraries were merged as the input of InfoGenomeR, and paired-end data were used for CN callings and allele-specific and haplotype estimation. SNPs were detected using BCFtools (version 1.3)[54]. Reads from three libraries of the HeLa transcriptome were mapped and quantified using HISAT2[55] and CuffLinks[56], and expression values were measured by collecting the mean counts from the duplicates. The WGS data of lung cancer cell lines, TCGA samples, and metastatic breast cancer included pre-processed data (BAM) mapped to GRCh37, and variants were called in the same way as the paired-end HeLa and simulated data sets.

**Reporting summary**. Further information on research design is available in the Nature Research Reporting Summary linked to this article.

## Data availability

WGS and RNA-seq data of the HeLa cell line are available in the database of Genotypes and Phenotypes (dbGaP; accession code No. phs000643.v10.p1). WGS data of lung cancer cell lines are available in the dbGaP (accession code phs000299.v2.p1). WGS data of TCGA samples are available in the dbGaP (accession code phs000178.v11.p8). WGS data of relapsed or metastatic breast cancers are available in the EGA (accession code EGAD00001002696). Simulated data sets from NA12878 for GRCh37 and GRCh38 are available in Zenodo [https://doi.org/10.5281/zenodo.4545666]. Simulated data sets from HG00732, NA19238, and HG00513 for GRCh37 are available in Zenodo [https://doi.org/10.5281/zenodo.4556315]. The remaining data are available within the Article and Supplementary Information, or from the authors upon request.

## Code availability

InfoGeomeR is implemented by R (version 3.4.3) and C++ (version 4.8.2), and available on GitHub at https://github.com/dmcblab/InfoGenomeR. The code for cancer genome simulation is available on GitHub at https://github.com/dmcblab/InfoGenomeR_simulation.

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

## Acknowledgements

This work was supported by Institute of Information & communications Technology Planning & Evaluation (IITP) grant funded by the Korea government (MSIT) (No. 2019-0-00567, Development of Intelligent SW systems for uncovering genetic variation and developing personalised medicine for cancer patients with unknown molecular genetic mechanisms) and a National Research Foundation of Korea (NRF) grant funded by the Korea government (NRF-2016R1A2B2013855).

## Author contributions

H.L. initiated and supervised the project, H.L. and Y.L. developed the algorithm, collected data, analysed the results and wrote the paper, and Y.L. performed experiments.

## Competing interests

The author declares no competing interests.
