## [Peer Review File · Nature Communications]

Reviewers' comments:

Reviewer #2 (Remarks to the Author): Expert in computational genomics

Lee & Lee address the problem of karyotype reconstruction in cancer by developing a computational method for creating putative karyotypic reconstructions with NGS data - using a combination of structural variant calls, ploidy and copy number variation data to create an allele-specific breakpoint graph. This method is introduced in a bioinformatic tool called "InfoGenomeR." InfoGenomeR uses an integer programming strategy to resolve an allele-specific breakpoint graph. To evaluate their tool, the authors ran InfoGenomeR on simulated data, data from the HeLa cell line and 174 cancer samples in TCGA.

By aggregating preexisting bioinformatic tools for SV and CN determination, the authors show increased F1 score of their tool's aggregated SV/CN calls on simulated data against SV/CN calls produced by individual tools. Furthermore they report the identification of SV clusters whose topology they assign based on graph structure (e.g. DM versus DM/HSR). They demonstrate that such SV clusters are sometimes shared between cancer metastases or are sometimes private.

The stated goal of this computational method is extraordinarily ambitious; to aggregate SV calls from NGS data into successful haplotype-resolved karyotypes in cancer. This particular goal is riddled with pitfalls such as the presence of tumor heterogeneity, the lack of existence of multiple validation datasets, and the broadly different classes of genomic rearrangements that can occur in tumors (DM, HSR, BFB, chromothripsis, etc.). Some of these pitfalls perhaps are reflected in the finding that in the HeLa data analyzed here, less than half of interchromosomal translocations (as predicted by m-FISH) were recapitulated by InfoGenomeR. Indeed, their paper falls short on many accounts:

The assertion (appearing multiple times in the paper), that InfoGenomeR is "a new method to reconstruct the complete cancer genomes [sic]" seems to be a gross overstatement of the tool's abilities. Outputting a haplotype-phased breakpoint graph based on data aligned to a reference genome does not, in this reviewer's opinion, constitute a complete reconstruction of a cancer genome. Moreover many cancer genomes have aneuploidies, focal amplifications and very complex rearrangements like chromothripsis, duplications, breakage fusion bridge, and others. As far as we can tell, the output of InfoGenomeR is certainly different from say, a complete de-novo assembly of a cancer genome. The authors can claim that by building haplotype resolved structures they can understand chains of structural variation in cancer, but not claim to assemble the cancer genome.

As is addressed briefly in the paper, InfoGenomeR does not attempt to resolve low-complexity regions. The bias introduced by alignment to reference could be quite large and remains unquantified in this paper. Please quantify this.

Given that InfoGenomeR establishes a haplotype graph, it is not clear the extent to which phasing errors exist in the output of InfoGenomeR. Ideally the authors would quantitatively demonstrate the rate of phasing errors in reconstructed haplotypes. While this is not necessarily a benchmark

possible with the TCGA dataset, they must show simulations that guide a reader through the strengths and weaknesses of their approach.

Even the inputs and output of their tool is not well defined. The authors must make the source code available for download on an open source repository, and provide some test input and output. In the paper, the authors should explicitly mention what is used as input and what exactly is generated as output.

What are the output files of InfoGenomeR and what do they indicate exactly. How much manual interpretation is required?

Do the authors intend to release an open-source tool?

Are ambiguous reconstructions of haplotype-resolved breakpoint graphs addressed in the outputs? It seems prudent that a comprehensive discussion of the ambiguities of InfoGenomeR reconstructions are also included.

What is the run-time of InfoGenomeR like? How does it scale with coverage?

This reviewer would also like to know how the choice of reference genome version affects the accuracy of the SV results in InfoGenomeR. Are significantly different breakpoint graphs detected between GRCh37 and GRCh38-aligned data? It appears GRCh37 was used throughout this paper. For the TCGA data in this paper, remapping to GRCh38 is of course unreasonable, but for simulated data it does seem like a valuable and vital comparison to make.

The classification of HSRs/DM status seems very powerful. Can the authors elaborate to what extent the tool reconstructs the exact structure of such focal amplifications? Or does it only classify the different topologies?

The authors describe in the discussion section that “Long-read sequencing is required to find SVs that cannot be identified by short-read sequencing. Our framework is likely to benefit from long-read sequencing technologies so that SV calls from short-read sets can be integrated.” Given that both long read datasets (e.g. from the Pacbio webset) and NGS data of publicly available cancer cell lines are available, do the authors think that this point might be immediately addressable to compare SVs detected in Pacbio data

versus by InfoGenomeR?

Detail points :

The authors describe a parameter lambda for local CN detection with BIC-seq2. Can the authors describe how the values were chosen and the rationale behind this, at least at a high-level?

How well does remapping of non-properly paired reads increase the ability to detect SV breakpoints? What proportion of SV edges were removed by the remapping? What proportion of SV edges were added by the remapping?

There should be a clearer distinction between “outperforming” a tool which is actually used in the InfoGenomeR pipeline and modifying the strategy for running that tool to get better results. For instance, for copy number estimation it was confusing the read that InfoGenomeR “displayed superior prediction over BIC-seq2”, when in fact InfoGenomeR uses BIC-seq2 to get its CN results, with a pre-segmented input. This strategy and distinction should be more clearly stated.

Tumor genome heterogeneity is a common and powerful driver of drug resistance in tumors. If presented with a sample in which one sub-population had one karyotype, and another sub-population had a different karyotype, would InfoGenomeR fail? Can the authors describe the level to which this heterogeneity is reconstructable with InfoGenomeR.

InfoGenomeR appears to provide valuable measurements of complex genomic rearrangements occurring throughout the cancer genome and represents a step forward in reconstruction of complex cancer genomes. However it is not quite clear that it rises to the lofty challenge of accurate karyotype reconstruction. As a result, perhaps a reframing of the stated goals to better reflect the most valuable output of this tool - the genome-wide breakpoint graph - would be a more accurate characterization of its abilities. Furthermore, the accuracy of haplotype phasing of the SVs (especially in the context of karyotypic abnormality) is not well quantified and needs to be more clearly addressed. Lastly, as a sincere suggestion, there are many places where the paper does not read well for grammatical reasons. We encourage the authors to review these issues carefully.

Reviewer #3 (Remarks to the Author): Expert in structural variation

In the presented study a novel computational framework/method InfoGenomeR for cancer genome's breakpoint graph reconstruction is introduced.

The main objective of the proposed method is to reconstruct a "balanced" copy-number (CN) annotated breakpoint graph that describes the sequenced cancer genome in question.

The manuscript is fairly written and is easy to follow for the people involved in the area of cancer genome's reconstruction, but can be a bit difficult for a broader audience.

The proposed theoretical framework seems somewhat derivative and more extensive discussion about the existing gaps in the area, that the proposed method addresses, is needed, alongside additional comparisons with other state-of-the-art tools (see major comments below).

Presented results seem to support the claim of the robust capability of the InfoGenomeR to infer rearranged cancer genome breakpoint graphs, though several design choices (e.g., breakpoint graph simplification; see major comments below) as well as the mentioned examples of missing translocation SVs (and most likely other) make the reader question the reliability of the presented results and its general applicability.

Overall, a major and comprehensive revision of the proposed manuscript is required.

Major comments:

* Comparison only with Weaver, the oldest method there is for the cancer genome graph reconstruction with notion of allele-specific CNs, is simply unacceptable, given that methods like ReMixt (McPherson et al 2017, "ReMixT: clone-specific genomic structure estimation in cancer") and JaBba (Hadi et al, 2019, "Novel patterns of complex structural variation revealed across thousands of cancer genome graphs") and others have been introduced with superior performance, when compared to Weaver. At least one of said tools need to be added to the comparison.

* It is unclear how much of the methodological framework is novel vs slightly reused versions of methods like ReMixT/JaBba/RCK/AmpliconArchitect. Theoretical description of the differences is needed to better illuminate the novelty of the methodological side of the presented manuscript.

* The usage of ensemble-based SV breakpoint detection with DELLY2, Manta, and novoBreak (or in multi-sample setting) is very vaguely described and requires additional information of the utilized/recommended settings for SV inference on the individual method level, as well as how the SVs across methods are merged into an input SV set.

* "Application of multi-sample WGS data to reveal tumor evolution" -- the benefit of the proposed method in this section is rather unclear, as the proposed algorithm only works with one genome at a time, thus suggesting separate applications to primary/metastasis samples is separate. In recent studies (e.g., Zaccaria and Rapahel, 2019 "Accurate quantification of copy-number aberrations and whole-genome duplications in multi-sample tumor sequencing data"), which is cited in the presented manuscript, it is shown that considering related clones (which cancer genomes in primary and metastasis are) in the allele-/haplotype-specific fashion separately can often lead to "allele/haplotype-flipping" of both segment and SV copy numbers as well as to reconstructions that outright contradict possible evolutionary history (e.g., segment present in a cloneB, but absent in cloneA, when cloneB is derived from cloneA). Furthermore, only SVs are considered for this multi-sample analysis, and not segment CNs, begging the question of why the usage of InfoGenomeR is even needed, when a simple SV inference/merging can do the job. An expanded discussion of the details of this analysis is needed. with its limitations clearly outlined for the reader.

* When talking about a balanced breakpoint graph a question of "unbalanced" extremities (i.e., telomeres) is mentioned, but we never see statistics over the total quantity of said (novel) telomeres in the recovered graphs. This is a rather important point, as authors do mention, that every telomere corresponds to a chromosome end, and statistics over the number of telomeres is crucial in understanding the rearranged structure of the underlying cancer genome.

* A notion of complete removal of germline variants when matching normal tissue sequencing data exists requires additional discussion as to why such removal would not affect the derived graph/karyotype organization when there are germline variations as compared to the reference genome. Same comment is applicable to the breakpoint graph simplification.

* A comparison of underlying theoretical notions between the Eulerian paths enumeration proposed in this study to the recent theoretical framework (Aganezov et al 2019, "Recovering rearranged cancer chromosomes from karyotype graphs") may be beneficial for the audience.

* While smaller SVs are excluded from Eulerian path enumeration, a theoretical proof/practical demonstration of how such removal does not affect the number of Eulerian decompositions is in order.

* Page 7: "However, InfoGenomeR could not identify 12 of the interchromosomal translocations from m-FISH, most of which were likely to be centromeric fusions that were not observed in WGS data, or occurred in repeat regions that were not mappable" -- after this statement a reader is left wondering how much the recovered Eulerian paths can be trusted, as every missing translocation would invalidate reported paths. More discussion is needed on this point.

* Complex rearrangement analysis seems to be very ad hoc and not grounded in any kind of evolutionary model, but rather based on the notion that "clustering SVs breakpoints indicate presence of complex rearrangements". More discussion needed on this point as well as comparison to methods like Shatterseek (Cortés-Ciriano et al 2020, "Comprehensive analysis of chromothripsis in 2,658 human cancers using whole-genome sequencing"), that have been designed for Chromothripsis identification.

* Software does not seem to be accessible and thus it is impossible to check its complexity in terms of installation/usage. This is paramount for reproducible science especially in the methodological manuscript like this one, where a method is presented for the audience with an aim of it being applied in various suitable research settings.

Minor comments:

* page 4: "...until the graph converges. The integer programming has two optimization schedules ..." -- very vague and unclear for the reader who is not deeply involved in the topic already. More context is needed.

* page 4: "Between two iterations, the reads which do not pair properly are remapped" -- no concepts that are referenced in this sentence are introduced at this points, making the sentence confusing.

* page 5: "...precision and recall rates in the total (0.987/0.825) and (0.981/0.919)..." -- style correction may be in order as currently prevision/recall values in parenthesis are not clearly attributed to precision and recall, respectively, but rather resemble fractions.

* page 6: "For integer CNs, Weaver could not detect total CNs." -- this is confusing, unclear, and somewhat misleading, as Weaver infers allele-specific CN profiles and obtaining a total CN value for every segment with inferred allele-specific CN states should be as simple as adding two allele-specific values together.

* page 8: "... we generated the karyotypic scenarios..." -- this suggests evolutionary history reconstruction, rather than derived karyotype structures

* page 14: "this graph is equivalent to the interval-adjacency graph" -- why the new terminology needed, is the graph is equivalent?

- * page 15: "estimated using ABSOLUTE" -- is InfoGenomeR capable of working with other CN estimation tools? What about allele-specific ones, like TitanCNA?
- * page 15: "segment of the node (s)" -- notation not introduced before and would benefit from an explicit introduction before usage.
- * page 15: "For high CNs that exceeded the ABSOLUTE estimation range" -- unclear how input CN values from ABSOLUTE exceed ABSOLUTE's estimation range. A clarification needed.
- * page 16: "performed integer programming" -- this needs a rewrite.
- * page 17: equation would benefit from explicit enumeration for future cross-references
- * page 18: "..where SV breakpoints were exactly reciprocal..." -- does this only correspond to full "cycles" (i.e., when for n SVs there are exactly $n/2$ pairs of reciprocal breakpoints), or any case of pairs of reciprocal breakpoints?
- * page 18: "... SVs with zero multiplicities are false positives and are removed iteratively" -- what does this mean? why is the removal (has to be?) iterative?
- * page 19: sentence that starts with "In addition, after breakpoint construction, SVs are classified..." -- this sentence is complex, confusing, and requires a rewrite.
- * page 20: "the number of heterozygous SNPs < 5" -- regardless of the size of the segment???
- * page 20: \bar{S} set is not explicitly defined
- * page 22, 25: equations would benefit from enumeration for future cross-reference
- * page 25/26: "Circular chromosomes including DMs were observed as cycles" -- how? isolated subgraphs with all vertices balanced, or some of them constituted parts of more complex subgraphs?
- * page 26: "... the number of candidate genome increases exponentially with the number of SV edges and edge multiplicities" -- this is rather inaccurate statement, as not every SV edge introduces a potential for multiple paths, and thus does not increase the overall path decomposition space.
- * page 26: Insertions SVs are defined here as operations often referred to as "block-interchange" in genome rearrangement literature, but it is unclear if regular insertion SVs (a single breakpoint addition of novel sequence) are supported.
- * page 26: "... there were n types of paths..." -- path types are never introduced
- * page 27: "breakpoint graph construction for multi-sample data" this paragraph is vague, with no methodological specifics and with usage of terminology, that has not been previously introduced (e.g., "...fine-tuning the shared and private..."). Requires a rewrite.
- * supplement, page 6: it seems that head-to-tail and tail-to-head are switched for tandem duplication and deletion SVs.
- * supplement, page 7: "removing the SVs affect ... and even lowers the objective function for the copy number balance conditions by removing copy number noises" -- if the short SV is a true one, would not the outline case constitute an error, rather than a simplification/noise removal?

Reviewer #2 (Remarks to the Author): Expert in computational genomics

Lee & Lee address the problem of karyotype reconstruction in cancer by developing a computational method for creating putative karyotypic reconstructions with NGS data - using a combination of structural variant calls, ploidy and copy number variation data to create an allele-specific breakpoint graph. This method is introduced in a bioinformatic tool called "InfoGenomeR." InfoGenomeR uses an integer programming strategy to resolve an allele-specific breakpoint graph. To evaluate their tool, the authors ran InfoGenomeR on simulated data, data from the HeLa cell line and 174 cancer samples in TCGA.

By aggregating preexisting bioinformatic tools for SV and CN determination, the authors show increased F1 score of their tool's aggregated SV/CN calls on simulated data against SV/CN calls produced by individual tools. Furthermore they report the identification of SV clusters whose topology they assign based on graph structure (e.g. DM versus DM/HSR). They demonstrate that such SV clusters are sometimes shared between cancer metastases or are sometimes private.

1. The stated goal of this computational method is extraordinarily ambitious; to aggregate SV calls from NGS data into successful haplotype-resolved karyotypes in cancer. This particular goal is riddled with pitfalls such as the presence of tumor heterogeneity, the lack of existence of multiple validation datasets, and the broadly different classes of genomic rearrangements that can occur in tumors (DM, HSR, BFB, chromothripsis, etc.). Some of these pitfalls perhaps are reflected in the finding that in the HeLa data analyzed here, less than half of interchromosomal translocations (as predicted by m-FISH) were recapitulated by InfoGenomeR. Indeed, their paper falls short on many accounts:

Answer) Thank you for the comment. In the revised manuscript, we clarified the goal of our study as reconstructing cancer genome karyotypes based on the haplotype graph construction. For validation, we added three validation datasets (A549, H226, and H229) with different classes of genomic rearrangements (balanced and unbalanced translocations, chromoplexy and chromothripsis), reconstruction of which have not been accomplished previously. In cell lines, most of SVs that were not recalled by InfoGenomeR are centromeric or telomeric fusions that are caused by technical limitation of WGS, not by algorithmic limitation. We found that the copy number breakpoints were found in the centromere or telomere regions (the reference sequence for them is "NNN..."). We could not find any read that was mapped on the regions.

We added three validation cell line datasets (A549, H226, and H229) to Figure 3 and new results were described in the Results section as follows. In addition, Table 1 and Supplementary Table S3 were newly added.

Pages 9-10) Validation using cancer cell lines. To evaluate the performance of InfoGenomeR, we analysed WGS data from three lung cancer cell lines (H292, A549, and H226)³¹ and the HeLa cell line³², whose karyotypes are well known. We constructed haplotype graphs for each cell line (Fig. 3). Because the graphs included multiple karyotypic possibilities as per the alternative Eulerian paths, we selected the one with the minimum entropy for the validation of karyotyping among the candidate karyotypes. The reconstructed karyotypes were matched with m-FISH karyotypes, and chromosomal ends predicted by InfoGenomeR were compared (Table 1 and Supplementary Table 3). InfoGenomeR identified 62.5%, 50.0%, 53.3%, and 40% of the interchromosomal translocations from m-FISH (Table 1). Most of the unidentified translocations were found in centromeric or telomeric regions (Supplementary Table 3). For the correctly identified interchromosomal translocations, InfoGenomeR can detect breakpoints at the base-pair resolution in the haplotype, and types of complex SVs, such as chromothripsis that cannot be revealed by m-FISH.

The H292 cell line showed chromoplexy (rearrangement chains)³³ among chromosomes 6, 8, 11, and 19 (T3-T6 and C2 in Fig. 3a), resulting in der(6)t(6;8), der(11)t(11;19) and der(19)t(11;19). The A549 cell line was triploid and showed chromothripsis in chromosomes 3 and 15 (C1 and C2, respectively, in Fig. 3b). We reconstructed der(19)t(15;19)x2 that was generated from chromothripsis of chromosome 15. In addition, we reconstructed the karyotype of the H226 cell line, which was tetraploid, with balanced translocations, t(8;19) and t(9;20) (T2-3 and T4-5, respectively, in Fig. 3c), and unbalanced translocations, t(7;10), t(10;15) and t(20;21) (T1, T6, and T7, respectively, in Fig. 3c). The derivative chromosomes were duplicated, which suggested that translocations were followed by whole-genome duplications (WGDs).

For the HeLa cell line, we identified nine translocations, of which, eight matched the translocations identified by m-FISH. The unmatched translocation was between 3p and a near-centromeric region of the chromosome, representing centromeric noise (T3 in Fig. 3d). Notably, we reconstructed representative

HeLa derivative chromosomes [der(1)t(1;3), der(12)t(3;12), and der(19)t(13;19)]³⁴ with InfoGenomeR at the base-pair resolution (Fig. 3d). Our results showed that chromosome 11 had the excessive SVs with the loss of heterozygosity, implicating that chromothripsis underlay der(11)t(7;11) (Fig. 3e).

- 2. The assertion (appearing multiple times in the paper), that InfoGenomeR is “a new method to reconstruct the complete cancer genomes [sic]” seems to be a gross overstatement of the tool’s abilities. Outputting a haplotype-phased breakpoint graph based on data aligned to a reference genome does not, in this reviewer’s opinion, constitute a complete reconstruction of a cancer genome. Moreover many cancer genomes have aneuploidies, focal amplifications and very complex rearrangements like chromothripsis, duplications, breakage fusion bridge, and others. As far as we can tell, the output of InfoGenomeR is certainly different from say, a complete de-novo assembly of a cancer genome. The authors can claim that by building haplotype resolved structures they can understand chains of structural variation in cancer, but not claim to assemble the cancer genome. As is addressed briefly in the paper, InfoGenomeR does not attempt to resolve low-complexity regions. The bias introduced by alignment to reference could be quite large and remains unquantified in this paper. Please quantify this.**

Answer) Thank you for the comment. We agree that “to reconstruct complete cancer genomes” may be considered overstatement. For example, we missed low-complexity regions, which can be reconstructed better using de-novo genome assembly methods. In this revision, we relaxed the term as “reconstruct genome karyotypes” instead of reconstructing the complete cancer genomes. Karyotype reconstruction is more focused on genome-wide view rather than local low-complexity regions. We revised the manuscript to make our goal clear and examples of some changes are given as follows.

Title) Integrative reconstruction of cancer genome karyotypes using InfoGenomeR

Page 3) In this study, we aimed to develop a new method to reconstruct cancer genome karyotypes, based on complex topology analysis and offer a haplotype graph-based representation.

Page 18) In summary, we developed a novel method to reconstruct cancer genome karyotypes and explored the karyotypes of complex SVs in three cancer types (BRCA, GBM, and OV) and multi-sample data with primary and metastatic cancer cells.

- 3. Given that InfoGenomeR establishes a haplotype graph, it is not clear the extent to which phasing errors exist in the output of InfoGenomeR. Ideally the authors would quantitatively demonstrate the rate of phasing errors in reconstructed haplotypes. While this is not necessarily a benchmark possible with the TCGA dataset, they must show simulations that guide a reader through the strengths and weaknesses of their approach.**

Answer) Thank you for the comment. When we estimated accuracy of phasing of InfoGenomeR and Weaver, InfoGenomeR showed better performance for haplotype estimation than Weaver. We added this result to Figure 2, and the paragraph explaining haplotype phasing accuracy was inserted to the main manuscript as follows.

Page 8) For haplotype estimation, we measured the switch error rate between the true and inferred haplotypes, based on the total or somatic breakpoint graph. InfoGenomeR showed error rates of 1.98% and 1.87% for the total and somatic mode respectively (15X) (Fig. 2), and the small decrease in the error rate for the somatic mode might have resulted from the higher accuracy of somatic ASCN estimation. InfoGenomeR showed better performance for haplotype estimation than Weaver, because it could benefit from the better ASCN estimation than Weaver.

- 4. Even the inputs and output of their tool is not well defined. The authors must make the source code available for download on an open source repository, and provide some test input and output. In the paper, the authors should explicitly mention what is used as input and what exactly is generated as output. What are the output files of InfoGenomeR and what do they indicate exactly. How much manual interpretation is required? Do the authors intend to release an open-source tool? Are ambiguous reconstructions of haplotype-resolved breakpoint graphs addressed in the outputs? It seems prudent that a comprehensive discussion of the ambiguities of InfoGenomeR reconstructions are also included. What is the run-time of InfoGenomeR like? How does it scale with coverage?**

Answer 4-1) We released InfoGenomeR at Github (<https://github.com/dmclab/InfoGenomeR>) with required

inputs and outputs with demos, which we used for simulation study and the cell line validation. We added input and output explicitly in main paper as follows.

Page 3) The input of InfoGenomeR is SV calls, unmapped reads, read-depth information, and SNPs that were detected from reads alignment. InfoGenomeR integrates cancer purity and ploidy, total CNAs, allele-specific CNAs, and haplotype information to identify the optimal breakpoint graph representing cancer genomes. InfoGenomeR identifies haplotype information in three steps: (1) creation of an initial breakpoint graph from initial SV calls and integer copy number (CN) estimation, (2) construction of an allele-specific breakpoint graph with allele-specific CN (ASCN) estimation, and (3) generation of a haplotype breakpoint graph with haplotype reconstruction by phasing SNPs. InfoGenomeR produces haplotype-resolved SV and CNA outputs with the haplotype graph. Finally, InfoGenomeR classifies rearrangement topologies and derives cancer genome karyotypes from the haplotype graphical output (Supplementary Fig. 1).

Page 4) Source codes of InfoGenomeR and simulation data are available at <https://github.com/dmclab/InfoGenomeR>.

Answer 4-2) Run time is proportional to iterations for breakpoint graph construction (mean time is half an hour for one iteration.) and each of allele-specific and haplotype graph construction takes an hour. For simulation data (Demo 1 on Github), it may take about 7 hours (5 iterations and allele-specific and haplotype graph construction). For cell lines (Demo 2 on Github) or TCGA data will take about a day (40 iterations for breakpoint graphs and allele-specific and haplotype graph construction).

5. **This reviewer would also like to know how the choice of reference genome version affects the accuracy of the SV results in InfoGenomeR. Are significantly different breakpoint graphs detected between GRCh37 and GRCh38-aligned data? It appears GRCh37 was used throughout this paper. For the TCGA data in this paper, remapping to GRCh38 is of course unreasonable, but for simulated data it does seem like a valuable and vital comparison to make.**

Answer) Thank you for the comment. To compare the accuracy of InfoGenomeR depending on reference genome version (GRCh37 vs GRCh38), we measured performances for GRCh38-based simulated data. Although the improvement of the GRCh38 reference could reduce performance gaps among variant calling methods, the high performance of InfoGenomeR was still valid for the GRCh38 reference. We added Supplementary Note, Supplementary Figure 6 and a paragraph in the main paper as follows.

Page 8) To compare the performance depending on the human reference genome versions, we evaluated InfoGenomeR against the five other tools using GRCh38-based simulated data sets (Supplementary Note). Performance gaps between SV callers were reduced compared with those of the GRCh37-based simulated data sets (Supplementary Fig. 6). This reduction in performance gaps might have resulted from the mappability improvement of GRCh38. InfoGenomeR and JaBbA for total SVs, and InfoGenomeR and Manta for somatic SVs exhibited the best performances in that order, respectively. InfoGenomeR and JaBbA had similar performances for total CNA breakpoint calling. Although, the mappability improvement in the GRCh38 reference could reduce performance gaps among the variant calling methods, the high performance of InfoGenomeR was still valid for the GRCh38 reference. Considering these results, InfoGenomeR outperformed the other variant-calling methods in all restricted variant-calling categories, for both the GRCh37 and GRCh38 references.

Supplementary Note, Page 21) To compare performance depending on the human reference genome versions, we generated three additional germline genomes from the NA12878 based on GRCh38. SNP and SV positions were lifted over using the ucsc liftOver tool from phase 3 of the 1000 Genomes Project. All the simulation schemes were same as those used in the GRCh37 simulation, and reads were mapped to the GRCh38 reference genome using BWA-MEM.

6. **The classification of HSRs/DM status seems very powerful. Can the authors elaborate to what extent the tool reconstructs the exact structure of such focal amplifications? Or does it only classify the different topologies?**

Answer) Thank you for the comment. An important characteristic of HSR/DM is that genomic regions are repeatedly duplicated. Our Eulerian path finding approach with the minimum entropy can find out the representative structure of such focal amplifications. We can classify different topologies and provide their representative structures. We added Supplementary Fig. 1 to clarify the topology output.

Page 3) InfoGenomeR classifies rearrangement topologies and derives cancer genome karyotypes from the haplotype graphical output (Supplementary Fig. 1)

7. The authors describe in the discussion section that “Long-read sequencing is required to find SVs that cannot be identified by short-read sequencing. Our framework is likely to benefit from long-read sequencing technologies so that SV calls from short-read sets can be integrated.” Given that both long read datasets (e.g. from the Pacbio webset) and NGS data of publicly available cancer cell lines are available, do the authors think that this point might be immediately addressable to compare SVs detected in Pacbio data versus by InfoGenomeR?

Answer) Applying our InfoGenomeR to PacBio data might require an additional effort to incorporate merits of long-read data. Although your comment is very useful, we would like to leave it for the future work.

Detail points:

8. The authors describe a parameter lambda for local CN detection with BIC-seq2. Can the authors describe how the values were chosen and the rationale behind this, at least at a high-level?

Answer) The parameter lambda is determined based on sample noises. For simulation, we selected lambda, which gives good result for a subset of simulation data. For cancer cell lines, their data is quite noisy and we selected the higher lambda than those for simulation (noisy data require large lambda). We added parameter values as follows in the manuscript.

Page 20) InfoGenomeR divided genomic regions using current SV breakpoints. Then, in the pre-divided regions, it performed local CN segmentation with BIC-seq2 with the main penalty parameter λ . InfoGenomeR used different λ parameter values for first and second iterations, where the second iterations used a higher penalty, thereby allowing the mis-segmented regions without SV evidence to be merged. In the present analysis, the parameters bin size = 100, initial $\lambda = 1$, and final $\lambda = 16$ were used for the simulated data. Cancer cell line data showed a higher noise level than the simulated data, and the parameters bin size= 100, initial $\lambda = 1$, and final $\lambda = 2000$ were used for cancer cell line data, where the reconstructed karyotypes were well matched with the m-FISH karyotypes. The same parameters used in cancer cell line data were used for TCGA and EGA data subsequently.

9. How well does remapping of non-properly paired reads increase the ability to detect SV breakpoints? What proportion of SV edges were removed by the remapping? What proportion of SV edges were added by the remapping?

Answer) Thank you for the comment. For SVs and CNA breakpoints, we measured the performance changes from first-round iterations, the intermediate step (remapping of non-properly paired reads), to second-round iterations. We added detailed performance in Supplementary Figure 4, and explained the results in the main manuscript as follows.

Page 6) In addition, InfoGenomeR remapped non-properly paired reads to unbalanced nodes to discover SVs at the intermediate step, which resulted in a 2.8% improvement in the recall rate for somatic SVs (Supplementary Fig. 4).

Page 7) Specifically, InfoGenomeR predetermined CNA breakpoints using initial SVs (first-round iterations), discovered CNA breakpoints where candidate SVs existed (the intermediate step), and reduced false breakpoints in segmented regions by increasing the segmentation parameter (second-round iterations) (Supplementary Fig. 4)

10. There should be a clearer distinction between “outperforming” a tool which is actually used in the InfoGenomeR pipeline and modifying the strategy for running that tool to get better results. For instance, for copy number estimation it was confusing the read that InfoGenomeR “displayed superior prediction over BIC-seq2”, when in fact InfoGenomeR uses BIC-seq2 to get its CN results, with a pre-segmented input. This strategy and distinction should be more clearly stated.

Answer) We agree with the reviewer that distinction between “outperforming” and “showing an enhanced performance” is required on the comparison analysis. DELLY2, Manta, and novoBreak were used for SV callings and BIC-seq2 and ABSOLUTE were used for CNA calling. Performances of the independent individual tools were enhanced when they were reused in the integrative strategy of InfoGenomeR. When

compared with the other tools such as CREST, Weaver, and CONSERTING, InfoGenomeR can be considered as outperforming them. We clarified this distinction at the comparison analysis as follows.

Page 6) Our results showed that the integrative strategy of InfoGenomeR imparted an enhanced performance over individual SV tools (DELLY2, Manta, and novoBreak), and InfoGenomeR outperformed the other graph SV caller, JaBbA.

Page 7) InfoGenomeR displayed superior prediction over BIC-seq2.

⇒ InfoGenomeR exhibited an enhanced performance over BIC-seq2.

Page 7) Among three methods, InfoGenomeR showed the best performance in detecting integer CNs.

⇒ InfoGenomeR showed an enhanced performance over the combination of BIC-seq2 and ABSOLUTE and outperformed JaBbA, achieving the best performance in detecting integer CNs.

- 11. Tumor genome heterogeneity is a common and powerful driver of drug resistance in tumors. If presented with a sample in which one sub-population had one karyotype, and another sub-population had a different karyotype, would InfoGenomeR fail? Can the authors describe the level to which this heterogeneity is reconstructable with InfoGenomeR.**

Answer) Thank you for the comment. Currently we focused on cancer cell lines (>90% purity and nonclonal) and TCGA samples with (>70% purity and nonclonal). We validated our framework for nonclonal cancer samples (Figure 3). We can construct the breakpoint graph for the subclone if we estimate subclonal copy numbers. For future work, we will expand our framework to reconstruct breakpoint graphs for subclones as we previously discussed in discussion.

- 12. InfoGenomeR appears to provide valuable measurements of complex genomic rearrangements occurring throughout the cancer genome and represents a step forward in reconstruction of complex cancer genomes. However it is not quite clear that it rises to the lofty challenge of accurate karyotype reconstruction. As a result, perhaps a reframing of the stated goals to better reflect the most valuable output of this tool - the genome-wide breakpoint graph - would be a more accurate characterization of its abilities. Furthermore, the accuracy of haplotype phasing of the SVs (especially in the context of karyotypic abnormality) is not well quantified and needs to be more clearly addressed. Lastly, as a sincere suggestion, there are many places where the paper does not read well for grammatical reasons. We encourage the authors to review these issues carefully.**

Answer) Thank you for the comment. As we answered in the comment #2, we changed the title and revised the manuscript to clearly state our goal. We also proofread the manuscript again by an English editor.

Reviewer #3 (Remarks to the Author): Expert in structural variation

In the presented study a novel computational framework/method InfoGenomeR for cancer genome's breakpoint graph reconstruction is introduced.

The main objective of the proposed method is to reconstruct a "balanced" copy-number (CN) annotated breakpoint graph that describes the sequenced cancer genome in question. The manuscript is fairly written and is easy to follow for the people involved in the area of cancer genome's reconstruction, but can be a bit difficult for a broader audience. The proposed theoretical framework seems somewhat derivative and more extensive discussion about the existing gaps in the area, that the proposed method addresses, is needed, alongside additional comparisons with other state-of-the-art tools (see major comments below).

Presented results seem to support the claim of the robust capability of the InfoGenomeR to infer rearranged cancer genome breakpoint graphs, though several design choices (e.g., breakpoint graph simplification; see major comments below) as well as the mentioned examples of missing translocation SVs (and most likely other) make the reader question the reliability of the presented results and its general applicability.

Overall, a major and comprehensive revision of the proposed manuscript is required.

1. Comparison only with Weaver, the oldest method there is for the cancer genome graph reconstruction with notion of allele-specific CNs, is simply unacceptable, given that methods like ReMixT (McPherson et al 2017, "ReMixT: clone-specific genomic structure estimation in cancer") and JaBba (Hadi et al, 2019, "Novel patterns of complex structural variation revealed across thousands of cancer genome graphs") and others have been introduced with superior performance, when compared to Weaver. At least one of said tools need to be added to the comparison.

Answer) Thank you for the comment. We agree that the recent graph-based SV callers, such as JaBba and ReMixT, are required to compare with InfoGenomeR. When we tested both JaBba and ReMixT, the performance of ReMixT was much lower than JaBba and it took long time to run ReMixT. In addition, the JaBba paper reported that JaBba outperformed ReMixT. Thus, for the full performance comparison, we compared JaBba and InfoGenomeR, and found that InfoGenomeR outperformed JaBba. In this revision, we added Supplementary Figure 2 for detailed comparison between JaBba and InfoGenomeR, and added the comparison between JaBba and InfoGenomeR throughout the "InfoGenomeR outperforms other variant-calling methods" section (page 5) in the main manuscript.

Pages 5-6) To compare InfoGenomeR with JaBba, which is the recent graph SV caller, we ran JaBba¹⁷ using the same SV union set input (DELLY2, Manta, and novoBreak) that we used for InfoGenomeR. Because JaBba was sensitive to the input purity and ploidy hyperparameter, we used the purity and ploidy estimation of InfoGenomeR for the JaBba input. We tested various hyperparameter settings for JaBba along with the JaBba recommendation, and selected the best setting for SV detection (Supplementary Fig. 2).

Page 6) Our results showed that the integrative strategy of InfoGenomeR showed an enhanced performance over individual SV tools (DELLY2, Manta, and novoBreak), and InfoGenomeR outperformed the other graph SV caller, JaBba.

Page 7) JaBba showed the second best performance for CNA breakpoints.

Page 7) JaBba was compared with InfoGenomeR for both total and somatic integer CNs. InfoGenomeR showed an enhanced performance over the combination of BIC-seq2 and ABSOLUTE and outperformed JaBba, achieving the best performance in detecting integer CNs.

2. It is unclear how much of the methodological framework is novel vs slightly reused versions of methods like ReMixT/JaBba/RCK/AmpliconArchitect. Theoretical description of the differences is needed to better illuminate the novelty of the methodological side of the presented manuscript

Answer) Thank you for the comment. We described the novelty of the InfoGenomeR method more clearly in the introduction and discussion sections as follows.

Pages 2-3) High-throughput sequencing has advanced our understanding of SVs by resolving the genomic changes at the single-base-level. Early-stage methods have been developed to detect SVs using discordant and split reads from sequencing data⁷⁻¹⁰; however, these methods are limited to detecting SV breakpoints in

local genomic windows. Recently, several advanced methods¹¹⁻²⁰ that integrate genomic information, such as cancer purity and ploidy, total CNAs, allele-specific CNAs, and haplotype information, have been developed to identify SVs. They use a graph-based representation for rearranged cancer genomes but do not analyse the actual karyotypes of linear and/or circular chromosomes; thus not producing karyotypic topologies such as HSRs, DMs, and chromothripsis. Global reconstruction of cancer genome karyotypes in cancers may allow us to uncover the mechanism of cancer development and evolution.

Pages 14-15) InfoGenomeR allows identification of complex rearrangement topologies (HSR, DM, HSR/DM, and CT) in the reconstructed cancer genome karyotypes. In a previous study, the identification of DM has been conducted using integrating SVs and CNAs¹³, but the analysis was restricted to local amplified regions without recovering haplotype karyotypes. ShatterSeek²⁰ used an integrative approach of SVs and CNAs to identify CT; however it did not provide karyotype structures such as derivative chromosomes and DMs resulted from CT. Recently, a decomposition method for DMs and/or linear chromosomes based on a haplotype graph has been introduced⁴⁶. Nevertheless, this method lacks interpretation of other topologies such as HSRs, HSR/DMs, or CTs. JaBbA¹⁷ introduced other complex topologies with DMs, except for karyotypes that were not derived from reconstructed haplotypes. InfoGenomeR enables us to understand complex topologies with karyotype reconstruction simultaneously at the genome-wide level, as shown in the analysis of TCGA (Fig. 4) and EGA data (Fig. 5). InfoGenomeR helps to identify, for the first time, the recurrent derivative chromosomes generated from chromosomes 11 and 17 with HSRs in BRCA. Our analysis of the SV clusters showed that CCND1 and ERBB2 were often closely clustered in these derivative chromosomes. Besides, we found that GBMs and OVAs were mainly characterized by HSR/DM or DM and HSR by fold-back inversions on different chromosomes.

- 3. The usage of ensemble-based SV breakpoint detection with DELLY2, Manta, and novoBreak (or in multi-sample setting) is very vaguely described and requires additional information of the utilized/recommended settings for SV inference on the individual method level, as well as how the SVs across methods are merged into an input SV set.**

Answer) Thank you for the comment. We added the detailed versions and commands in Supplementary Table 4, and added the explanation how to merge SV calling sets in the Methods section (page 19).

Page 19) Initial SV detection by InfoGenomeR. The variant callers DELLY2⁷, Manta⁸, and novoBreak⁹ were used with default parameters to detect initial SVs with or without controls (total or somatic). Low-quality SVs, defined as <3 variant supporting reads or a mapping quality <20, were filtered out. Breakpoints of an SV were sorted by the chromosomal and coordinate order in the reference sequence, and the SV was annotated as head-to-head (HH), head-to-tail (HT), tail-to-head (TH), or tail-to-tail (TT) depending on the orientation of breakpoint adjacencies to the genomic segments. The head and tail are the 5' and 3' coordinates in the reference genome, respectively. Detailed settings for SV detection were shown in Supplementary Table 4. SV sets from individual SV callers were unified as the input of InfoGenomeR. If breakpoints predicted by the SV callers could differ for the same SV (if breakpoints of SVs were overlapped in <100bp, they were considered as the same SV), we empirically selected one of their breakpoints when SV sets were unified.

- 4. "Application of multi-sample WGS data to reveal tumor evolution" -- the benefit of the proposed method in this section is rather unclear, as the proposed algorithm only works with one genome at a time, thus suggesting separate applications to primary/metastasis samples is separate. In recent studies (e.g., Zaccaria and Rapahel, 2019 "Accurate quantification of copy-number aberrations and whole-genome duplications in multi-sample tumor sequencing data"), which is cited in the presented manuscript, it is shown that considering related clones (which cancer genomes in primary and metastasis are) in the allele/haplotype-specific fashion separately can often lead to "allele/haplotype-flipping" of both segment and SV copy numbers as well as to reconstructions that outright contradict possible evolutionary history (e.g., segment present in a cloneB, but absent in cloneA, when cloneB is derived from cloneA). Furthermore, only SVs are considered for this multi-sample analysis, and not segment CNs, begging the question of why the usage of InfoGenomeR is even needed, when a simple SV inference/merging can do the job. An expanded discussion of the details of this analysis is needed. with its limitations clearly outlined for the reader.**

Answer) Thank you for the comment. We added discussion as follows.

Pages 16-17) Through a multi-sample analysis, we could identify the evolutionary processes of HSR and DM generation with CTs, during metastatic tumour evolution. Previously, SVs have been investigated to emerge in metastasis. However, the discrimination between private and shared SVs was unclear, and karyotypic characterisation has not been performed. We observed that SVs could be misidentified as private SVs by a simple SV calling approach, even though they were shared SVs with supporting CNA evidence

existing in primary and metastatic tumour. We performed imputation for candidate shared SVs that may exist in both primary and metastatic tumour during breakpoint graph construction, thus clearly distinguished true private SVs. These private SVs were shown together in HSR and DM topology with CT (Fig. 5d and 5e). We characterised their karyotypes by reconstructing derivative chromosomes and DMs, thus providing a basis for structure-based analyses of tumour evolution. Nevertheless, there were limitations in the current analysis. First, our applications to the primary and metastatic tumours were independent with each other even though we adjusted breakpoint graphs in the intermediate steps during iterative optimisations. In addition, we did not perform clone-specific interpretation, although sub-clonal SVs or CNAs may clarify tumour evolution processes. A joint approach for subclones across multi-samples⁴⁷ will be required for future analyses.

5. **When talking about a balanced breakpoint graph a question of "unbalanced" extremities (i.e., telomeres) is mentioned, but we never see statistics over the total quantity of said (novel) telomeres in the recovered graphs. This is a rather important point, as authors do mention, that every telomere corresponds to a chromosome end, and statistics over the number of telomeres is crucial in understanding the rearranged structure of the underlying cancer genome.**

Answer) Thank you for the comment. In this revision, we reconstructed three additional cell lines, and their predicted karyotypes were shown in Figure 3 in addition to the HeLa cell line. Thus, to show novel telomeres in the recovered graph, we specified chromosomal ends for the four cancer cell lines in Supplementary Table S3 along with their reconstructed karyotypes.

6. **A notion of complete removal of germline variants when matching normal tissue sequencing data exists requires additional discussion as to why such removal would not affect the derived graph/karyotype organization when there are germline variations as compared to the reference genome. Same comment is applicable to the breakpoint graph simplification.**

Answer) Thank you for the comment. A reason for removing germline variants and short simple SVs is that they do not have enough allele-specific and haplotype information because of their short lengths. Another reason is that germline variants caused over-segmentation of a genome that prevented reconstruction of the chromosomal karyotypes. So, removal of them was a design choice to obtain chromosomal karyotypes in the more genome-wide view, rather than considering small SVs with insufficient allele-specific information in local regions. We assumed that such removal was negligible in the karyotyping view (such germline variants will generate small "transits" rather than change the entire path) and detailed consideration for germline variants was left for the future work. We added discussion as follows.

Page 7 in the Supplementary information) Breakpoint graph simplification Short simple SVs (<100 kb) are negligible in a karyotypic scope, and the number of heterozygous SNPs in the short segment could be insufficient to provide confident allelic ratios, which are important for allele-specific graph construction. Thus, after breakpoint graph construction, we exclude short simple SVs classified as tandem duplications, deletions, insertions, or inversions, and then subtract copy number bins in short SVs. Removing these SVs and copy number bins affects few Eulerian paths in a karyotypic scope, while maintaining the path orientation. In addition, short simple SVs caused an oversegmentation of the genome that prevented an accurate measurement of multiplicities of segment edges across the genome. For instance, the true measurement of segment edges for ABC and ABBCx2 triploid genome is $u(A) = 3$, $u(B) = 5$, $u(C) = 3$, respectively. However, an error of the segment edge multiplicity for C could arise as $u(C) = 4$. By removing the B segment and representing the segment as A[removed B]C, we could measure $u(A[removed B]C) = 3$ accurately, which lowers the objective function for the copy number balance condition. Here, we replace ABC and ABBCx2 genome with ABCx3 genome as the simplification process when B is short. The simplification requires additional iterative steps until the breakpoint graph converges after removing SV edges and copy number bins for simple SVs.

7. **A comparison of underlying theoretical notions between the Eulerian paths enumeration proposed in this study to the recent theoretical framework (Aganezov et al 2019, "Recovering rearranged cancer chromosomes from karyotype graphs") may be beneficial for the audience.**

Answer) Thank you for the comment. We formulated the reconstruction of cancer genome as minimum-entropy Eulerian path finding problem while Aganezov et al 2019 formulated it as a Eulerian decomposition problem that describes the most parsimonious and biologically relevant scenario. We cited the Aganezov paper and described our method in more detail in the Methods section.

Pages 31-32) To identify the candidate genomes, Eulerian paths were enumerated to alternate between segment edges and SV/reference edges on the haplotype graph constructed in the previous step. Head and tail nodes that did not satisfy the copy number balance condition (including original telomere ends) were considered as ends of the reconstructed chromosomes P, which could also be true ends or breaks due to missing SVs or miscalculated CNs. Circular chromosomes, C, included in the DM cluster were observed as circular paths. Eulerian decomposition problem (EDP) was defined to find linear and circular chromosomes from the breakpoint graph⁴⁶. Although the min-EDP, which minimized the number of paths and cycles, $|P|+|C|$, was previously suggested to describe the most possible karyotype⁴⁶, the min-EDP was not always biologically relevant (i.e. the max-EDP could be the case). In this study, we formulated minimum-entropy Eulerian path enumeration that prioritized the decomposition of P and C with the minimum entropy. To enumerate candidate Eulerian paths, a multi-decision DAG efficient in enumerating Eulerian paths on the multigraph was used. The multi-decision DAG was expanded in a root-to-leaves model by sequentially increasing the level and processing of each node in the breakpoint graph (Supplementary Methods and Supplementary Fig. 18). Leaf nodes represented possible edge-pairing states delineating Eulerian paths reaching every genomic segment.

8. **Page 7: "However, InfoGenomeR could not identify 12 of the interchromosomal translocations from m-FISH, most of which were likely to be centromeric fusions that were not observed in WGS data, or occurred in repeat regions that were not mappable" -- after this statement a reader is left wondering how much the recovered Eulerian paths can be trusted, as every missing translocation would invalidate reported paths. More discussion is needed on this point.**

Answer) Thank you for the comment. We validated our method using three more cell lines, and the reconstructed karyotypes were shown in Figure 3. In addition, we showed the missing translocations in Table 1 and Supplementary Table S3. In Supplementary Table S3, we examined the detailed missing translocations, and found that most missing translocations are located in the centromere, which shows that these centromeric fusions cannot be identified using WGS data.

We added three validation cell line datasets (A549, H226, and H229) to Figure 3 and new results were described in the Results section as follows. In addition, Table 1 and Supplementary Table S3 were newly added.

Pages 9-10) Validation using cancer cell lines. To evaluate the performance of InfoGenomeR, we analysed WGS data from three lung cancer cell lines (H292, A549, and H226)³¹ and the HeLa cell line³², whose karyotypes are well known. We constructed haplotype graphs for each cell line (Fig. 3). Because the graphs included multiple karyotypic possibilities as per the alternative Eulerian paths, we selected the one with the minimum entropy for the validation of karyotyping among the candidate karyotypes. The reconstructed karyotypes were matched with m-FISH karyotypes, and chromosomal ends predicted by InfoGenomeR were compared (Table 1 and Supplementary Table 3). InfoGenomeR identified 62.5%, 50.0%, 53.3%, and 40% of the interchromosomal translocations from m-FISH (Table 1). Most of the unidentified translocations were found in centromeric or telomeric regions (Supplementary Table 3). For the correctly identified interchromosomal translocations, InfoGenomeR can detect breakpoints at the base-pair resolution in the haplotype, and types of complex SVs, such as chromothripsis that cannot be revealed by m-FISH.

The H292 cell line showed chromoplexy (rearrangement chains)³³ among chromosomes 6, 8, 11, and 19 (T3-T6 and C2 in Fig. 3a), resulting in $der(6)t(6;8)$, $der(11)t(11;19)$ and $der(19)t(11;19)$. The A549 cell line was triploid and showed chromothripsis in chromosomes 3 and 15 (C1 and C2, respectively, in Fig. 3b). We reconstructed $der(19)t(15;19)x2$ that was generated from chromothripsis of chromosome 15. In addition, we reconstructed the karyotype of the H226 cell line, which was tetraploid, with balanced translocations, $t(8;19)$ and $t(9;20)$ (T2-3 and T4-5, respectively, in Fig. 3c), and unbalanced translocations, $t(7;10)$, $t(10;15)$ and $t(20;21)$ (T1, T6, and T7, respectively, in Fig. 3c). The derivative chromosomes were duplicated, which suggested that translocations were followed by whole-genome duplications (WGDs).

For the HeLa cell line, we identified nine translocations, of which, eight matched the translocations identified by m-FISH. The unmatched translocation was between 3p and a near-centromeric region of the chromosome, representing centromeric noise (T3 in Fig. 3d). Notably, we reconstructed representative HeLa derivative chromosomes [$der(1)t(1;3)$, $der(12)t(3;12)$, and $der(19)t(13;19)$]³⁴ with InfoGenomeR at the base-pair resolution (Fig. 3d). Our results showed that chromosome 11 had the excessive SVs with the loss of heterozygosity, implicating that chromothripsis underlay $der(11)t(7;11)$ (Fig. 3e).

9. **Complex rearrangement analysis seems to be very ad hoc and not grounded in any kind of evolutionary model, but rather based on the notion that "clustering SVs breakpoints indicate**

presence of complex rearrangements". More discussion needed on this point as well as comparison to methods like Shatterseek (Cortés-Ciriano et al 2020, "Comprehensive analysis of chromothripsis in 2,658 human cancers using whole-genome sequencing"), that have been designed for Chromothripsis identification.

Answer) Thank you for the comment. We agree with the reviewer that complex rearrangement analysis was not based on the evolutionary model. However, our analysis can characterize complex events (HSR and DM) related with CTs in the karyotype level. We discussed our result by comparing with the ShatterSeek method in the discussion section as follows.

Page 15-16) CTs were recently reported to be found in more than half of cancers by ShatterSeek, where CTs with other complex events were more prevalent than canonical CTs that showed an oscillating pattern between two CN states²⁰. However, the goal of ShatterSeek was restricted to figure out SV clusters of CTs, and the structures of derivative chromosomes were not investigated comprehensively because of the lack of a reconstruction strategy. Our results showed HSR, HSR/DM, or DM topologies involved with CT in chromosomal structures by reconstructing cancer genome karyotypes. We found that chromosome 17 is a template chromosome, which was recurrently rearranged with other chromosomes with CTs in BRCA, demonstrating that complex events with CTs in multiple chromosomes generated derivative chromosomes. It was suggested that complex events involved with CTs in the formation of derivative chromosomes contributed to the amplification of cancer-related genes, such as CCND1, CDK4, MDM2, and ERBB2²⁰. Our results showed that cancer-related genes were amplified in the formation of derivative chromosomes. Taken together, we provided new insights into the karyotypic view of complex rearrangements involved with CTs.

- 10. Software does not seem to be accessible and thus it is impossible to check its complexity in terms of installation/usage. This is paramount for reproducible science especially in the methodological manuscript like this one, where a method is presented for the audience with an aim of it being applied in various suitable research settings.**

Answer) We released the source code of InforGenomeR at <https://github.com/dmclab/InfoGenomeR>.

Page 4) Source codes of InfoGenomeR and simulation data are available at <https://github.com/dmclab/InfoGenomeR>.

Minor comments:

- 11. page 4: "...until the graph converges. The integer programming has two optimization schedules ..." -- very vague and unclear for the reader who is not deeply involved in the topic already. More context is needed.**
- 12. page 4: "Between two iterations, the reads which do not pair properly are remapped" -- no concepts that are referenced in this sentence are introduced at this points, making the sentence confusing.**

Answers to comments 11-12) Thank you for the comment. We changed as follows to make the meaning clear.

Pages 4-5) First, InfoGenomeR evaluates all reads in WGS datasets, and generates initial SV calls using the tools DELLY²⁷, Manta⁸, and novoBreak⁹ (Fig. 1a), and performs initial CN segmentation using BIC-seq2²⁶. Then, it constructs an initial breakpoint graph of local genomic segments using the initial SV and CN breakpoints. The breakpoint graph is composed of nodes and segment edges, reference edges, and SV edges. The following three-step iterations update the initial breakpoint graph. In each iteration, i) local genomic segments are refined, ii) integer CNs of genomic segments are estimated using purity and ploidy (ABSOLUTE²⁷), and iii) the integer programming of the CN balance condition²⁸ determines the edge multiplicities of the breakpoint graph and removes zero-multiplicity SVs. Each iteration restarts with the SV set without zero-multiplicity SVs, CN segmentation is performed without the previous false-positive SV breakpoints, and integer CNs of segments are recalculated. Iterations are performed until the graph converges (no zero-multiplicity SV is observed). The iterations are composed of first and second rounds of iterations depending on the segmentation parameter, and the CN segments are merged with their neighbour CN segments more commonly in the second-round iterations than in the first-round iterations. At the intermediate step between the first and second rounds of iterations, the discordant or unmapped reads, which do not pair properly, are remapped to the sequences of candidate adjacencies from unbalanced nodes. (Fig. 1b). Then, candidate adjacencies supported by their reads are generated, and the second-round iterations finalise the breakpoint graph. Next, integer CNs are divided into ASCNs using negative binomial

models for the different depths of heterozygous SNPs, and the expectation-maximisation (EM) algorithm is used for estimating parameters. Integer programming under the CN balance condition with the ASCNs constructs the allele-specific breakpoint graph and then the imbalanced heterozygous SNP sequences are phased (Fig. 1c). Genomic segments with balanced heterozygous SNPs are phased using a hidden Markov model (BEAGLE²⁹), and the final haplotype breakpoint graph is constructed (Fig. 1d). Eulerian paths can be enumerated to obtain candidate genomes by pairing breakpoint graph edges in a multi-decision directed acyclic graph (DAG)³⁰. In the end, InfoGenomeR generates candidate karyotypes of the cancer cells at the haplotype level (Fig. 1e).

13. **page 5: "...precision and recall rates in the total (0.987/0.825) and (0.981/0.919)..." -- style correction may be in order as currently prevision/recall values in parenthesis are not clearly attributed to precision and recall, respectively, but rather resemble fractions.**

Answer) Thank you for the comment. We changed as follows to make the meaning clear.

Pages 6) InfoGenomeR achieved the highest total (precision, 0.987; recall, 0.825) and somatic (precision, 0.981; recall, 0.919) SV calling performance, at a haplotype coverage of 15X

14. **page 6: "For integer CNs, Weaver could not detect total CNs." -- this is confusing, unclear, and somewhat misleading, as Weaver infers allele-specific CN profiles and obtaining a total CN value for every segment with inferred allele-specific CN states should be as simple as adding two allele-specific values together.**

Answer) Thank you for the comment. We edited the sentence to prevent misleading.

Page 7) Weaver was able to detect total integer CNs, but not somatic integer CNs with the germline coverage control.

15. **page 8: "... we generated the karyotypic scenarios..." -- this suggests evolutionary history reconstruction, rather than derived karyotype structures**

Answer) Thank you for the comment. We edited the sentence to prevent misleading.

Page 11) we derived the karyotype structures

16. **page 14: "this graph is equivalent to the interval-adjacency graph" -- why the new terminology needed, is the graph is equivalent?**

Answer) Thank you for the comment. We removed the unnecessary sentence.

17. **page 15: "estimated using ABSOLUTE" -- is InfoGenomeR capable of working with other CN estimation tools? What about allele-specific ones, like TitanCNA?**

Answer) Thank you for the comment. Currently, InfoGenomeR is not working with other CN estimation tools. We will consider it in future.

18. **page 15: "segment of the node (s)" -- notation not introduced before and would benefit from an explicit introduction before usage.**

Answer) Thank you for the comment. We added the explanation as follows.

Page 20) The end sides of a genomic segment were represented by a head and tail node, and the copy ratio and integer CN of the genomic segment of the node s (head or tail) were denoted by copy ratio(s) and CN(s) respectively.

19. **page 15: "For high CNs that exceeded the ABSOLUTE estimation range" -- unclear how input CN values from ABSOLUTE exceed ABSOLUTE's estimation range. A clarification needed.**

Answer) Thank you for the comment. We clarified the sentence.

Page 21) Because ABSOLUTE assigns the predefined maximum integer CN when the copy ratio(s) is larger than the estimation limit, in this case, we calculated non-integer CN'(s) satisfying the copy ratio equation,

copy ratio(s) = $\{CN'(s)p+2(1-p)\}/D$, and then, CN(s) is assigned as $[CN'(s)]$, rounding of the non-integer $CN'(s)$.

20. page 16: "performed integer programming" -- this needs a rewrite.

Answer) Thank you for the comment. We clarified the sentence.

Page 22) ... solved the integer programming problem (Eq. 2) to find the multiplicities of edges

21. page 17: equation would benefit from explicit enumeration for future cross-references.

Answer) Thank you for the comment. We added explicit enumeration.

22. page 18: "...where SV breakpoints were exactly reciprocal..." -- does this only correspond to full "cycles" (i.e., when for n SVs there are exactly n/2 pairs of reciprocal breakpoints), or any case of pairs of reciprocal breakpoints?

Answer) The explanation corresponds to any cases of pairs of reciprocal breakpoints.

23. page 18: "... SVs with zero multiplicities are false positives and are removed iteratively" -- what does this mean? why is the removal (has to be?) iterative?

Answer) Thank you for the comment. We clarified the sentence as follows. As you commented, removal itself is not iterative, and SVs with zero multiplicities were removed per each iteration.

Page 24) Notably, SVs with zero multiplicities are false positives and are removed before the next iteration.

24. page 19: sentence that starts with "In addition, after breakpoint construction, SVs are classified..." -- this sentence is complex, confusing, and requires a rewrite.

Answer) Thank you for the comment. We rewrote the sentences.

Pages 24-25) In addition, after breakpoint construction, SVs are classified as simple or complex SVs, based on the breakpoint graph (Supplementary Methods). Germline variants and short simple SVs (<100kb) are bottlenecks for karyotype reconstruction, because they do not have sufficient allelic information for the allele specific graph and may cause an over-segmentation of the genome. Assuming that they are negligible in the karyotyping view, we simplify the breakpoint graph by removing SV edges and copy number bins for germline variants (Supplementary Methods and Supplementary Fig. 15)

25. page 20: "the number of heterozygous SNPs < 5" -- regardless of the size of the segment???

Answer) Thank you for the comment. Currently, we used this condition regardless of the size of the segment.

26. page 20: \bar{S} set is not explicitly defined

Answer) Thank you for the comment. We missed definition of \bar{S} and edited the sentence as follows.

Page 26) $AG(S;E)$ was constructed, where the node set $S = \bar{S} \cup S1 \cup S2$ was composed of balanced (\bar{S}) and imbalanced nodes ($S1$ and $S2$ for temporal two haplotypes).

27. page 22, 25: equations would benefit from enumeration for future cross-reference

Answer) Thank you for the comment. We added explicit enumeration.

28. page 25/26: "Circular chromosomes including DMs were observed as cycles" -- how? isolated subgraphs with all vertices balanced, or some of them constituted parts of more complex subgraphs?

Answer) Thank you for the comment. The DM cluster is a complex subgraph, and it contains circular paths (all vertices of the subgraph of circular paths is balanced). We edited the sentence to prevent misleading.

Page 31) Circular chromosomes, C , included in the DM cluster were observed as circular paths.

29. page 26: "... the number of candidate genome increases exponentially with the number of SV edges and edge multiplicities" -- this is rather inaccurate statement, as not every SV edge introduces a potential for multiple paths, and thus does not increase the overall path decomposition space

Answer) Thank you for the comment. We agree that it is could be inaccurate. Thus, we revised the sentence

as follows.

Page 32) The enumeration for all the Eulerian paths is an NP-hard problem, and we prioritized Eulerian paths with the minimum entropy as biologically relevant cases.

30. **page 26: Insertions SVs are defined here as operations often referred to as "block-interchange" in genome rearrangement literature, but it is unclear if regular insertion SVs (a single breakpoint addition of novel sequence) are supported.**

Answer) Thank you for the comment. The regular insertion SVs are not supported. Thus, we edited the sentence as follows.

Page 32) tandem duplications, deletions, and block-interchange insertions

Page 6 in the Supplementary information) we classify SVs as simple SVs (tandem duplications, deletions, inversions, or block-interchange insertions) and complex SVs

Page 6 in the Supplementary information) Block-interchange insertions (INS): The segment is flanked by SVs and inserted to other chromosomal regions

page 26: "... there were n types of paths..." -- path types are never introduced

Answer) Thank you for the comment. We added explanation and edited the sentence.

Page 32) a node in the multi-decision DAG at level l represented edge-pairing states of a total of l node in the breakpoint graph, and there were n subsets of paths (the duplicated paths constitute a subset). The total number of paths was $w = w_1 + w_2 + \dots + w_n$, where w_i is the number of duplicated paths of the i^{th} subset. The entropy at level l , e_l , was derived from the following formula:

31. **page 27: "breakpoint graph construction for multi-sample data" this paragraph is vague, with no methodological specifics and with usage of terminology, that has not been previously introduced (e.g., "...fine-tuning the shared and private..."). Requires a rewrite.**

Answer) Thank you for the comment. We edited the paragraph in the Method section and described the detailed procedure in the Supplementary Note.

Page 33) Breakpoint graph construction for multi-sample data. For multi-samples, we unified SV sets from the initial SVs of each sample. Using the unified SV sets, the breakpoint graph of each sample was constructed. Then, we classified SVs into private and shared SVs depending on the existence of raw SV evidence (discordant or split reads) (Supplementary Table 5). If a shared SV is not called in a primary tumour, it is observed as a private SV in a metastatic tumour, and vice versa. This approach using the unified SV sets has an advantage that uncalled SV edges in each sample can be added to the graph if there are supporting copy number depths and adjacent SV information in that sample. For the cases with difficulties in distinguishing private and shared SVs, we made another round of iterative optimisation (Supplementary Note).

32. **supplement, page 6: it seems that head-to-tail and tail-to-head are switched for tandem duplication and deletion SVs.**

Answer) Thank you for the comment. The annotation is based on the reference sequence, so a head-to-tail SV corresponds to tandem duplication. We added additional explanation for the annotation.

Page 19) Breakpoints of an SV were sorted by the chromosomal and coordinate order in the reference sequence, and the SV was annotated as head-to-head (HH), head-to-tail (HT), tail-to-head (TH), or tail-to-tail (TT) depending on the orientation of breakpoint adjacencies to the genomic segments.

Page 6 in the Supplementary information) Note that head-to-head, head-to-tail, tail-to-head, and tail-to-tail notations are assigned based on the orientation of the breakpoint adjacencies to the genomic segments in the coordinate order of the reference sequence.

33. **supplement, page 7: "removing the SVs affect ... and even lowers the objective function for the copy number balance conditions by removing copy number noises" -- if the short SV is a true one, would not the outline case constitute an error, rather than a simplification/noise removal?**

Answer) Thank you for the comment. We added discussion in the Supplementary information.

Page 7 in the Supplementary information) Breakpoint graph simplification

Short simple SVs (<100 kb) are negligible in a karyotypic scope, and the number of heterozygous SNPs in the short segment could be insufficient to provide confident allelic ratios, which are important for allele-specific graph construction. Therefore, after breakpoint graph construction, we exclude short simple SVs, classified as tandem duplications, deletions, insertions, or inversions, and then subtract copy number bins in short SVs. Removing these SVs and copy number bins affects few Eulerian paths in a karyotypic scope, while still maintaining the path orientation. In addition, short simple SVs caused an oversegmentation of the genome that prevented an accurate measurement of multiplicities of segment edges across the genome. For instance, the true measurement of segment edges for the ABC and ABBCx2 triploid genomes is $u(A) = 3$, $u(B) = 5$, $u(C) = 3$, respectively. However, an error in the segment edge multiplicity for C could arise as $u(C) = 4$. By removing the B segment and representing the segment as A[removed B]C, we could measure $u(A[removed B]C) = 3$ accurately, thereby lowering the objective function for the copy number balance condition. Here, we replace the ABC and ABBCx2 genomes with the ABCx3 genome for the simplification process. The simplification requires additional iterative steps until the breakpoint graph converges after removing the SV edges and copy number bins for simple SVs.

REVIEWERS' COMMENTS

Reviewer #2 (Remarks to the Author):

The authors have significantly improved the manuscript and have addressed most of my concerns. In summary, their method InforGemomeR is an addition to the wgs based tools for cancer genome elucidation. It is one of the only tools for karyotyping, including the prediction of derivative chromosomes through translocation events, as also DM, HSR formation and chromothripsis while also incorporating SV calling and allele specific copy number calling. In the absence of gold-standard, it is hard to test the method but the authors have done a good job of comparing against m-FISH derived cancer karyotypes.

On the negative side, the description of the method is still not very good. A lot of terms are introduced with no prior definitions, making it very difficult to read the paper properly. These are described as minor comments below, but the authors must do a serious rewrite of the paper. They could also use a better equation editor.

Minor comments:

=====

Page 11: In HSR/Dm designation, what does the phrase "five repetitive cycle mean?" Do you mean a cycle with CN=5 throughout?

Page 19: The partitioning of the edge set into three classes is unclear. Specifically, are there only two node classifications: 'head' and 'tail,' and if so, what is the difference between segment edge and reference edge? Please describe this better.

Page 20: Please explain the parameter lambda. The reader at this point has no idea what lambda=1, 16, and 2000 mean, and how were these values chosen?

Page 20: what does "copy ratio" mean? Also, what is 'm' in "with a mean copy ratio $m_q = \{qp + 2(1 - p)\}/D$."

Page 21. 's' corresponds to both a segment-edge and a node leading to poor notation such as $e_s(s)$. Instead, can you use v for node throughout?

Page 25: "Let $A = \{A_1, A_2, \dots, A_{\lfloor (\mu(es(s))+1)/2 \rfloor}\}$ indicate a set of $\lfloor (\mu(es(s)) + 1)/2 \rfloor$ cases that the integer CN can be divided into"

This sentence is impossible to understand. What are cases? What does it mean to divide the CN into some number of cases? The A-i's are also not defined. One suspects that they are related to allele frequencies, but some information should be provided.

Page 26: "an allele-specific break-point graph $AG(S, E)$ was constructed, where the node set $S = S \cup$

S1 U S2 was composed of balanced (S) and imbalanced nodes (S1 and S2 for temporal two haplotypes)"

Again, I could not find the definition of balanced and unbalanced nodes. What does "temporal two haplotypes" mean?

Reviewer #3 (Remarks to the Author):

In the presented review authors have greatly improved the quality of the submitted manuscript as well as the underlying research study. Majority of the my concerns were either addressed or discussion points have been added.

Addition of comparisons with JaBba brings a much needed comprehensive nature to the evaluation process. Clarification on the not fully resolved linear/circular structures of the cancer chromosomes is also helpful to the reader as to properly measure expectations for the results from InfoGenomeR.

Additional information about recovered karyotypes and information about the observed novel telomeres is very useful. If time allows, a more general graphical representation of simple chromosomal cnts between true karyotypes and the obtained graph-based ones can be helpful to the reader.

Additional grammar/punctuation proofreading.

Ensure in all equations text is in different font vs math notation/symbols.

Reviewer #2 (Remarks to the Author):

The authors have significantly improved the manuscript and have addressed most of my concerns. In summary, their method InforGemomeR is an addition to the wgs based tools for cancer genome elucidation. It is one of the only tools for karyotyping, including the prediction of derivative chromosomes through translocation events, as also DM, HSR formation and chromothripsis while also incorporating SV calling and allele specific copy number calling. In the absence of gold-standard, it is hard to test the method but the authors have done a good job of comparing against m-FISH derived cancer karyotypes.

On the negative side, the description of the method is still not very good. A lot of terms are introduced with no prior definitions, making it very difficult to read the paper properly. These are described as minor comments below, but the authors must do a serious rewrite of the paper. They could also use a better equation editor.

Answer) We sincerely appreciate the positive evaluation of our study. According to the comments, we revised the description of the method and added explanations.

Minor comments:

1. **Page 11:** In HSR/Dm designation, what does the phrase "five repetitive cycle mean?" Do you mean a cycle with CN=5 throughout?

Answer) Thank you for the comment. The five repetitive cycles mean a cycle with five multiplicities. It could cause misleading, so we changed the phrase in the main manuscript and the Supplementary Information.

Page 11) HSR/DM (an SV cluster with high amplification connected to a chromosomal arm and at least five repetitive cycles)

⇒ HSR/DM (an SV cluster with high amplification connected to a chromosomal arm and a cycle with at least five multiplicities)

Page 11) DM (an SV cluster with at least five repetitive cycles, without any connection to a chromosomal arm).

⇒ DM (an SV cluster with a cycle with at least five multiplicities, without any connection to a chromosomal arm).

Supplement, page 7) HSR/DM: The cluster satisfies the HSR condition, and at least five repetitive cycles exist in the SV cluster from the minimum entropy search

⇒ HSR/DM: The cluster satisfies the HSR condition, and a cycle with at least five multiplicities exists in the SV cluster from the minimum entropy search

Supplement, page 8) DM: At least five repetitive cycles exist in the SV cluster from the minimum entropy search without being connected to a chromosomal arm.

⇒ A cycle with at least five multiplicities exists in the SV cluster from the minimum entropy search without being connected to a chromosomal arm.

2. **Page 19:** The partitioning of the edge set into three classes is unclear. Specifically, are there only two node classifications: 'head' and 'tail,' and if so, what is the difference between segment edge and reference edge? Please describe this better.

Answer) Thank you for the comment. For clarity, we added more description about segment and reference edges with head and tail node notations in the main manuscript.

Pages 19-21) InfoGenomeR constructs a breakpoint multigraph $G(S,E)$ from genomic segments and SVs. A node set S has two types, head nodes (Sh) and tail nodes (St), representing the head and tail sides of genomic segments, respectively, In the breakpoint graph, the i th genome segment is represented by a pair of the head

and tail node, (s_h^i, s_t^i) . An edge set (E) has three types: segment edge (E_s), reference edge (E_r), and SV edge (E_v). The segment edge connects the head node (s_h^i) and tail node (s_t^i) of i th genomic segment, and the multiplicity of the segment edge represents the CN of the genomic segment. The reference edge connects the tail node (s_t^i) and the head node (s_h^{i+1}) between i th and $i + 1$ th segment, representing the adjacency between adjacent genomic segments present in the reference genome. Conversely, the SV edge represents a novel adjacency between genomic segments that do not exist in the reference genome. The following iterative procedures construct the breakpoint graph:

3. Please explain the parameter lambda. The reader at this point has no idea what lambda=1, 16, and 2000 mean, and how were these values chosen?

Answer) Thank you for the comment. The lambda parameter is explained in the BIC-seq2 paper that we cite, but it seems to be necessary to explain the lambda in the text to understand the method. We added brief explanation of BIC-seq2 in the main manuscript.

Page 20) InfoGenomeR divided the genomic regions using current SV breakpoints. Then, in the pre-divided regions, it performed local CN segmentation with BIC-seq2²⁴ with the main penalty parameter λ , and measured the copy ratio between observed and expected read counts (from a control, if available) in the genomic regions. Briefly, BIC-seq2 uses the Bayesian information criterion to determine breakpoints that are composed of two terms²⁴: the negative log likelihood term, which explains how well the model with the breakpoint fits the read depth data, and the penalty term, which is proportional to the number of breakpoints and prevents over-segmentation. The parameter λ adjusts the penalty term, with higher λ preventing excessive breakpoints.

4. Page 20: what does "copy ratio" mean? Also, what is 'm' in "with a mean copy ratio $m_q = \{qp + 2(1 - p)\}/D$."

Answer) Thank you for the comment. The copy ratio means a ratio between observed read counts and expected read counts (from a control, if available) in the genomic regions. We added the sentence in the main manuscript.

m_q is a typo of \underline{m}_q , which denotes a mean copy ratio for the integer copy number q . We edited it.

Page 20) it performed local CN segmentation with BIC-seq2²⁴ with the main penalty parameter λ , and measured the copy ratio between observed and expected read counts (from a control, if available) in the genomic regions.

Page 21) The copy ratio(s) was fitted with a Gaussian mixture model, each component of which was a Gaussian distribution representing the integer CN state (q) with a mean copy ratio, $\underline{m}_q = \{qp + 2(1 - p)\}/D$.

5. Page 21. 's' corresponds to both a segment-edge and a node leading to poor notation such as $e_s(s)$. Instead, can you use v for node throughout?

Answer) Thank you for the comment. We intended the genomic 's'egment is represented by 's' rather than (v_h^i, v_t^i) using the v notation, which may help understand the node s meaning. In addition, notations like copy ratio(s) and CN(s) intended that s indicates the node s for the 's'egment, which may help understand the notations better than using copy ratio(v) and CN(v). We added the following sentence in the main manuscript rather than changing v to s .

Page 19) In the breakpoint graph, the i th genome segment is represented by a pair of the head and tail node, (s_h^i, s_t^i) .

6. "Let $A = \{A_1, A_2, \dots, A[(\mu(es(s))+1)/2]\}$ indicate a set of $[(\mu(es(s)) + 1)/2]$ cases that the integer CN can be divided into" This sentence is impossible to understand. What are cases? What does it mean to divide the CN into some number of cases? The A_i 's are also not defined. One suspects that they are related to allele frequencies, but some information should be provided.

Answer) Thank you for the comment. We added the following description for the notation and an example for A_i in the main manuscript.

Page 25) In addition to integer CNs based on total read depths, read depths of heterozygous SNPs provide

information about allele-specific CN. The integer CN, $\mu(e_s(s))$ of each segment from the breakpoint graph is divided into allele-specific CNs, ASCN(s), using heterozygous SNPs (if a control exists, all the heterozygous SNPs in the control are used). Let $A = \{A_1; A_2, \dots, A_{\lfloor (\mu(e_s(s))+1)/2 \rfloor}\}$ denote all the possible states of allele-specific CNs that the genomic segment can have, where the integer CN can be divided into a set of $\lfloor (\mu(e_s(s))+1)/2 \rfloor$ possible cases, and for each $A_i = \{A_{i,1}, A_{i,2}\}$, $A_{i,1} + A_{i,2} = \mu(e_s(s))$. For example, if the multiplicity of the segment edge, $\mu(e_s(s)) = 3$, there are two cases, $A_1 = \{0, 3\}$ and $A_2 = \{1, 2\}$.

Page 26: "an allele-specific break-point graph $AG(S, E)$ was constructed, where the node set $S = S \cup S_1 \cup S_2$ was composed of balanced (S) and imbalanced nodes (S_1 and S_2 for temporal two haplotypes)"

Again, I could not find the definition of balanced and unbalanced nodes. What does "temporal two haplotypes" mean?

Answer) Thank you for the comment. We added the definition of the balanced and unbalanced nodes and more description for temporal haplotype meaning in the main manuscript.

Pages 26-27) Based on the ASCNs, an allele-specific breakpoint graph $AG(S;E)$ was constructed, where the node set $S = \bar{S} \cup S_1 \cup S_2$ was composed of balanced (S) and imbalanced nodes (S_1 and S_2 for temporal two haplotypes), which denote the heads and tails of genomic segments with balanced and imbalanced ASCNs, respectively. In the allele-specific breakpoint graph, the imbalanced nodes are assigned to haplotype 1 or haplotype 2 temporally, whereas the balanced nodes are not assigned. The phased states (haplotype 1 and haplotype 2) of imbalanced nodes are preserved within the imbalanced ASCNs and can be switched across genomic segments with balanced ASCNs.

Reviewer #3 (Remarks to the Author):

In the presented review authors have greatly improved the quality of the submitted manuscript as well as the underlying research study. Majority of the my concerns were either addressed or discussion points have been added.

Addition of comparisons with JaBba brings a much needed comprehensive nature to the evaluation process. Clarification on the not fully resolved linear/circular structures of the cancer chromosomes is also helpful to the reader as to properly measure expectations for the results from InfoGenomeR.

Answer) We sincerely appreciate the positive evaluation of our study. According to the comments, we added a supplementary figure and conducted proofreading of the manuscript.

Additional information about recovered karyotypes and information about the observed novel telomeres is very useful. It time allows, a more general graphical representation of simple chromosomal cnts between true karyotypes and the obtained graph-based ones can be helpful to the reader.

Answer) Thank you for the comment. We agree that karyotype information described in Supplementary Table 3 can be difficult to read. We added Supplementary Fig. 7 that describes karyotypes and novel telomeres between true and recovered karyotypes in a graphical representation.

Additional grammar/punctuation proofreading.

Ensure in all equations text is in different font vs math notation/symbols.

Answer) Thank you for the comment. We carefully conducted proofreading carefully again, and we changed texts (non-italic in a roman font) in all equations to be distinguishable from math symbols (italic).